# Comparative physicochemical characterization and sensory profiling of Western Algerian and Polish honeys

Dalila Bereksi-Reguig[1], Hocine Allali[1]*, Salim Bouchentouf[2,3], Nessrine Kazi Tani[1,4], Grażyna Kowalska[5], Dariusz Kowalczyk[6], Jakub Wyrostek[7], Ewelina Zielińska[7], Radosław Kowalski[7]*

1 Department of Chemistry, Faculty of Sciences, Abou Bekr Belkaïd University, Tlemcen, Algeria, 2 Laboratory of Natural and Bioactive Substances (LASNABIO), Department of Chemistry, Faculty of Sciences, Abou Bekr Belkaïd University, Tlemcen, Algeria, 3 Doctor Tahar Moulay University of Saida, Saïda, Algeria, 4 Laboratory of Application of Electrolytes and Organic Poly-electrolytes (LCPO), Abou Bekr Belkaïd University, Tlemcen, Algeria, 5 Department of Tourism and Recreation, University of Life Sciences in Lublin, Lublin, Poland, 6 Department of Biochemistry and Food Chemistry, University of Life Sciences in Lublin, Lublin, Poland, 7 Department of Analysis and Evaluation of Food Quality, University of Life Sciences in Lublin, Lublin, Poland

* hocine.allali@univ-tlemcen.dz (HA); radoslaw.kowalski@up.lublin.pl (RK)

## Abstract

### Background

An in-depth analysis was conducted on 37 honey samples from western Algeria representing diverse floral sources—lavender, rosemary, sweet white mustard, thyme, milk thistle, carob, orange, euphorbia, eucalyptus, camphor, jujube, sage, harmal, and multifloral blends. The objective was to evaluate their physicochemical properties and sensory characteristics, with Polish honeys serving as references.

### Methods

Key physicochemical traits were measured, including moisture, pH, free acidity, electrical conductivity, hydroxymethylfurfural (HMF), proline, specific optical rotation, sugar profile (fructose, glucose, sucrose), and colour in CIELAB space (L*, a*, b*, $C_{ab}^{*}$ and $h_{ab}^{\circ}$). Sensory evaluation was performed using a five-point hedonic scale (+2 = "like very much" to –2 = "dislike very much") and an 11-descriptor Check-All-That-Apply (CATA) questionnaire.

### Results

All values satisfied European quality limits (moisture 14.67–20.87%, pH 3.47–5.60, free acidity 8.00–40.33 meq/kg, conductivity 0.16–1.18 mS/cm, HMF 1.79–49.43 mg/kg, sucrose < 5 g/100 g, proline 265.95–1200.66 mg/kg). Polish samples scored higher for taste (+1.22 ± 0.42 vs +0.18 ± 0.52; p = 0.009) and aroma (+0.72 ± 0.43 vs

**Data availability statement:** All raw data (physicochemical measurements, sensory scores, CATA frequencies), the Excel workbooks with summary tables and formulas, and the SPSS syntax used for statistical analyses are freely available from the Zenodo repository (https://doi.org/10.5281/zenodo.16386576).

**Funding:** This work was conducted as part of project No. B00L01UN130120220004, funded by the Ministry of Higher Education and Scientific Research of the People's Democratic Republic of Algeria and Abou BekrBelkaïd University, Tlemcen, Algeria. This work was financed by a statutory activity subsidy from the Polish Ministry of Science and Higher Education for the Faculty of Food Science and Biotechnology of the University of Life Sciences in Lublin. The APC was financed by a statutory activity subsidy from the Polish Ministry of Education and Science for the Faculty of Food Science and Biotechnology and for the Faculty Agrobioengineering of the University of Life Sciences in Lublin.

**Competing interests:** The authors have declared that no competing interests exist.

−0.26 ± 0.36; p = 0.016), whereas colour did not differ (p = 0.459). CATA indicated Algerian honeys were chiefly "mild" and "herbal", contrasting with Polish "sweet" and "sharp" profiles. Principal-component analysis (PC1 + PC2 ≈ 65% variance) and hierarchical clustering defined three groups: (A) sweet aromatic (all Polish + four Algerian), (B) moderately mild, and (C) sharp–bitter–herbal. Selected Algerian varietals—rosemary (S29) and multifloral (S14)—matched Polish hedonic acceptance, highlighting their premium/exotic market potential.

## Conclusion

Western Algerian honeys exhibit high compositional quality and distinctive sensory signatures, supporting competitiveness in food and health applications.

---

## 1. Introduction

Honey is a natural product containing more than 200 substances, including sugars, proteins, amino acids, water, enzymes, organic acids, vitamins, minerals, phenolic compounds, and volatile constituents [1–3]. It is widely used as both a food and a natural sweetener without additives, with a long shelf life and suitability for culinary applications [4]. The composition of honey varies according to its botanical and geographical origins, which influence its physicochemical parameters such as colour, moisture, and acidity, and consequently its organoleptic properties [5,6].

According to the Codex Alimentarius, honey is defined as "the natural sweet substance produced by honey bees from the nectar of plants, from secretions of living parts of plants, or from excretions of plant-sucking insects" [7,8]. Beyond its nutritional and functional roles, honey has socioeconomic significance through beekeeping, which contributes to pollination, agricultural economies, rural development, and biodiversity conservation [9]. Bees and hive products are also increasingly recognised as bioindicators of environmental pollution [10].

Honeys are typically classified as monofloral or multifloral depending on whether they originate predominantly from one floral source or from several. Multifloral honeys, also known as "polyfloral" or "all-flower" honeys, result from the contribution of multiple plant species within a given region and season [11]. Increasing consumer demand for authenticated honeys has stimulated interest in characterising both monofloral and multifloral varieties [12,13].

The botanical origin of honey is primarily determined through melissopalynology, the analysis of pollen grains [14]. This is often complemented by physicochemical and sensory analyses to provide a more comprehensive characterisation [15]. Additionally, volatile compound profiling has emerged as a reliable and precise technique for verifying authenticity [16]. By integrating these methods, a robust and definitive framework is established for authenticating honey and its floral provenance. This multi-faceted approach ensures a complete and accurate determination of the product's origin.

In Algeria, systematic data on beekeeping and honey production remain scarce. The available evidence suggests that the sector comprises approximately 1.2 million

colonies and 20,000 beekeepers, with a substantial increase in honey production between 2002 and 2010. However, the mean yield per hive remains below 4 kg [9]. Western Algeria, with its Mediterranean–Saharan climate and diverse flora, offers conditions conducive to the production of honeys with distinctive physicochemical and sensory profiles.

Poland represents an established European honey market with well-documented physicochemical and sensory standards. Comparative analyses between Algerian and Polish honeys are therefore relevant for assessing quality and positioning Algerian varieties in international contexts. Recent studies have also shown that declared geographical origin influences consumer perception and willingness to pay more strongly than intrinsic taste attributes [17], underscoring the importance of linking physicochemical data with sensory evaluation.

This study aimed to characterise the physicochemical parameters (moisture, pH, free acidity, electrical conductivity, hydroxymethylfurfural content, proline, specific optical rotation, sugar composition, and colour) of 37 honey samples from western Algeria of diverse botanical origins, and to evaluate their sensory properties in comparison with four Polish reference honeys. The overarching goal was to assess the quality profiles of Algerian honeys and their potential positioning relative to established European references.

## 2. Materials and methods

### 2.1 Chemicals and reagents

The chemicals employed comprised hydroxymethylfurfural (HMF, ≥ 99%) and proline, both supplied by Sigma-Aldrich® (Chemie GmbH, Taufkirchen, Germany). Ninhydrin, ethylene glycol monomethyl ether, and 2-propanol were obtained from Sigma-Aldrich® (Darmstadt, Germany). All reagents were of analytical-grade purity.

### 2.2 Honey samples

Thirty-seven honey samples, identified by codes S1–S37, were collected from experienced beekeepers across different regions of western Algeria, including Tlemcen, Ain-Temouchent, Sidi Bel Abbes, Mostaganem, Mascara, Tiaret, Naâma, and Bechar (Fig 1). The botanical sources of these samples included lavender (*Lavandula vera* D.C.), rosemary (*Rosmarinus officinalis* L.), sweet white mustard (*Sinapis alba* L.), thyme (*Thymus vulgaris* L.), milk thistle (*Silybum marianum* (L.) Gaertn.), carob (*Ceratonia siliqua* L.), orange (*Citrus sinensis* L.), euphorbia (*Euphorbia* L.), eucalyptus (*Eucalyptus globules* Labill.), camphor tree (*Cinnamomum camphora* L.), jujube (*Ziziphus lotus* L.), sage (*Salvia officinalis* L.), harmal (*Peganum harmala* L.), and multifloral honey. Table 1 provides detailed information on the type, region (GPS coordinates, climate), and botanical origins of the honey samples, including the scientific and common names of the plants that constitute their basic flora. All samples were collected between March 2017 and August 2018 and stored in sealed amber glass containers at 4 °C, protected from light and humidity. Previous studies have shown that honey remains stable under such conditions [18], and the samples exhibited no signs of fermentation, crystallisation, or sensory defects at the time of evaluation in 2025.

Additionally, in the sensory evaluation, four Polish honeys were used as reference samples (C38: multifloral honey; C39: multifloral honey; C40: heather honey; C41: buckwheat honey). The Polish honeys were purchased from a local health food store and held veterinary certification permitting sale (Producer: Pasieka Michał Rombel, Mińsk Mazowiecki, Poland). Parallel physicochemical analyses of Polish references were not feasible; literature values were used to provide a broader representative range (see Table 5 in section 3.).

### 2.3 Moisture content

The refractometric method was employed to determine the water content of each honey sample using an Abbe 5 refractometer (Bellingham + Stanley Ltd., Xylem UK). Optical refractive index measurements were corrected to a standard temperature of 20°C by applying a correction coefficient of 0.00023/°C. All measurements were performed in triplicate, and the results were compared against the Chataway table [19], which provides the moisture content of honey in percentage.



**Fig 1. Map of western Algeria depicting the distribution of the studied honey samples.**

## 2.4 pH and free acidity

The pH of each honey sample was measured using an Adwa AD8000 pH meter (Szeged, Hungary). To determine free acidity, 10 g of honey was dissolved in 75 mL of distilled water in a 250 mL beaker, and the pH was measured by immersing the pH electrode into the solution. Free acidity was then determined by titrating this solution with 0.1 M NaOH until a pH of 8.3 was reached. Phenolphthalein was used as an indicator to detect the appearance of a persistent pink colour lasting approximately ten seconds. A blank test was performed using distilled water under the same conditions. All experiments were conducted in triplicate, and the results were expressed in milliequivalents (meq) per kilogram of honey, following the methods described by Bogdanov *et al.* [19].



**Table 1. Geographical origins of honey samples from western regions in Algeria.**

| Region | Sample | Flower type | Scientific name | Botanical family | Location | GPS coordinates | Climate | Altitude (m) | Harvest season/year |
|---|---|---|---|---|---|---|---|---|---|
| Tlemcen | S1 | Lavender | *Lavandula vera* D.C. | Lamiaceae | Sidi Djillali | 34° 28' 00'' N 1° 34' 60'' W | Subhumid | 1470 | Summer 2018 |
| | S2 | Rosemary | *Rosmarinus officinalis* L. | Lamiaceae | Sidi Djillali | 34° 28' 00'' N 1° 34' 60'' W | Subhumid | 1470 | Spring 2018 |
| | S3 | Multifloral | Multifloral | – | Sidi Djillali | 34° 28' 00'' N 1° 34' 60'' W | Subhumid | 1470 | Spring 2018 |
| | S4 | Multifloral | Multifloral | – | Sidi Djillali | 34° 28' 00'' N 1° 35' 00'' W | Subhumid | 1425 | Summer 2017 |
| | S5 | Multifloral | Multifloral | – | El Aricha | 34° 13' 22'' N 1° 15' 21'' W | Subhumid | 1270 | Summer 2017 |
| | S6 | Sweet white mustard | *Sinapis alba* L. | Brassicaceae | Aïn Fezza | 34° 52' 45'' N 1° 14' 18'' W | Subhumid | 846 | Summer 2017 |
| | S7 | Thyme | *Thymus vulgaris* L. | Lamiaceae | Beni Snous | 34° 38' 35'' N 1° 33' 41'' W | Subhumid | 835 | Spring 2018 |
| | S8 | Milk thistle | *Silybum marianum* (L.) Gaertn. | Asteraceae | Beni Snous | 34° 38' 35'' N 1° 33' 41'' W | Subhumid | 835 | Summer 2018 |
| | S9 | Multifloral | Multifloral | – | Oued Chouly | 34° 56' 52'' N 1° 03' 17'' W | Subhumid | 705 | Autumn 2017 |
| | S10 | Carob | *Ceratonia siliqua* L. | Fabaceae | Oued Chouly | 34° 56' 52'' N 1° 03' 17'' N | Subhumid | 705 | Autumn 2017 |
| | S11 | Thyme | *Thymus vulgaris* L. | Lamiaceae | Beni Mester | 34° 52' 00'' N 1° 25' 00'' W | Subhumid | 697 | Spring 2017 |
| | S12 | Carob | *Ceratonia siliqua* L. | Fabaceae | Béni Ghazli | 34° 52' 34'' N 1° 07' 56'' W | Subhumid | 624 | Spring 2017 |
| | S13 | Multifloral | Multifloral | – | Oued es Safsâf | 34° 55' 60'' N 1° 18' 00'' W | Subhumid | 551 | Summer 2018 |
| | S14 | Multifloral | Multifloral | – | Sebaa Chioukh | 35° 09' 50'' N 1° 21' 27'' W | Subhumid | 514 | Spring 2017 |
| | S15 | Multifloral | Multifloral | – | Hennaya | 34° 57' 00'' N 1° 22' 00'' W | Subhumid | 429 | Summer 2017 |
| | S16 | Orange | *Citrus sinensis* L. | Rutaceae | Remchi | 35° 03' 00'' N 1° 26' 00'' W | Subhumid | 213 | Spring 2017 |
| | S17 | Multifloral | Multifloral | – | Honaïne | 35° 10' 35'' N 1° 39' 18'' W | Subhumid | 197 | Spring 2018 |
| | S18 | Milk thistle | *Silybum marianum* (L.) Gaertn. | Asteraceae | Honaïne | 35° 10' 35'' N 1° 39' 18'' W | Subhumid | 197 | Summer 2018 |
| Ain-Temouchent | S19 | Multifloral | Multifloral | – | Oulhaça El Gherarba | 35° 13' 00'' N 1° 31' 00'' W | Subhumid | 232 | Spring 2018 |
| | S20 | Multifloral | Multifloral | – | Beni Ghanem | 35° 15' 16'' N 1° 25' 38'' W | Subhumid | 220 | Summer 2018 |
| | S21 | Multifloral | Multifloral | – | Bouzedjar | 35° 34' 28'' N 1° 10' 01'' W | Subhumid | 104 | Spring 2018 |
| Sidi Bel Abbes | S22 | Euphorbia | *Euphorbia* L. | Euphorbiaceae | Ras El Ma | 34° 29' 51'' N 0° 49' 10'' W | Semi-arid | 1105 | Spring 2017 |
| | S23 | Milk thistle | *Silybum marianum* (L.) Gaertn. | Asteraceae | Telagh | 34° 47' 06'' N 0° 32' 40'' W | Semi-arid | 987 | Spring 2017 |
| | S24 | Multifloral | Multifloral | – | Lamtâr | 35° 04' 14'' N 0° 47' 53'' W | Semi-arid | 578 | Spring 2017 |
| | S25 | Eucalyptus | *Eucalyptus globulus* Labill. | Myrtaceae | Sidi Brahim | 35° 15' 38'' N 0° 34' 03'' W | Semi-arid | 432 | Spring 2017 |

*(Continued)*

**Table 1.** (Continued)

| Region | Sample | Flower type | Scientific name | Botanical family | Location | GPS coordinates | Climate | Altitude (m) | Harvest season/year |
|---|---|---|---|---|---|---|---|---|---|
| Mostaganem | S26 | Camphor | *Cinnamomum camphora* L. | Lauraceae | Sidi Ali | 36° 06' 17" N 0° 25' 24" E | Semi-arid | 216 | Autumn 2017 |
| | S27 | Eucalyptus | *Eucalyptus globulus* Labill. | Myrtaceae | Mostaganem | 35° 56' 00" N 0° 05' 00" E | Semi-arid | 104 | Summer 2017 |
| | S28 | Orange | *Citrus sinensis* L. | Rutaceae | Bouguirat | 35° 45' 05" N 0° 15' 12" E | Semi-arid | 66 | Spring 2017 |
| Mascara | S29 | Rosemary | *Rosmarinus offici-nalis* L. | Lamiaceae | Djebel Stamboul | 35° 23' 00" N 0° 09' 00" E | Semi-arid | 932 | Spring 2017 |
| Tiaret | S30 | Multifloral | Multifloral | – | Tiaret | 34° 55' 00" N 1° 34' 60" E | Semi-arid | 1189 | Spring 2018 |
| Naâma | S31 | Multifloral | Multifloral | – | Aïn Sefra | 32° 45' 20" N 0° 35' 09" W | Arid | 1073 | Spring 2017 |
| | S32 | Jujube | *Ziziphus lotus* L. | Rhamnaceae | Aïn Sefra | 32° 45' 20" N 0° 35' 09" W | Arid | 1073 | Spring 2017 |
| | S33 | Jujube | *Ziziphus lotus* L. | Rhamnaceae | Aïn Ben Khelil | 33° 17' 25" N 0° 45' 51" W | Arid | 1156 | Spring 2017 |
| | S34 | Sage | *Salvia officinalis* L. | Lamiaceae | Naâma | 33° 17' 25" N 0° 45' 51" W | Arid | 1031 | Spring 2017 |
| | S35 | Harmal | *Peganum harmala* L. | Zygophylla-ceae | Mecheria | 33° 33' 00" N 0° 17' 00" W | Arid | 891 | Spring 2017 |
| Bechar | S36 | Multifloral | Multifloral | – | Djebel Antar | 31° 56' 34" N 1° 55' 52" W | Arid | 1953 | Winter 2017 |
| | S37 | Sweet white mustard | *Sinapis alba* L. | Brassicaceae | Oued Zouzfana | 32° 04 '01" N 1° 14' 27" W | Arid | 830 | Spring 2017 |

## 2.5 Hydroxymethylfurfural content (HMF)

The HMF content was measured using a SHIMADZU UVmini-1240 spectrophotometer (Shimadzu Europa GmbH, Germany), following the method described by White (1979). This procedure involved measuring the absorbance of a clarified solution containing 10% (w/v) honey dissolved in distilled water, compared to a reference solution of the same honey in which the chromophore at λmax (284 nm) of HMF was destroyed by adding 0.1% sodium bisulfite. The HMF content was expressed in mg/kg of honey [19].

## 2.6 Electrical conductivity (EC)

Electrical conductivity was measured at 20.0°C using a WTW inoLab® Cond conductivity meter (WTW GmbH & Co. KG, Weilheim, Germany). The measurements were performed on a solution containing 20% (w/v) honey dissolved in demineralised distilled water. Results were expressed in millisiemens per centimetre (mS/cm) [19].

## 2.7 Specific rotation

The specific rotation ($[\alpha]_D^{20}$) or optical activity was measured at 20°C using an OPTIKA polarimeter (Science Italy). Measurements were conducted in a clear solution containing 10% (w/v) honey dissolved in distilled water. This parameter is associated with the carbohydrate content of the honey [19].



## 2.8 Proline content

Proline, the main amino acid in honey, was quantified spectrophotometrically [12,20]. One milliliter of formic acid and 1 mL of a 3% ninhydrin solution prepared in ethylene glycol monomethyl ether were added to three test tubes containing 0.5 mL of deionized water (blank), 0.5 mL of a 32 mg/L proline standard solution (standard), and 0.5 mL of a 5% honey dilution (sample). After the tubes were sealed, the contents were mixed thoroughly and heated in a boiling water bath for 15 min, followed by 10 min in a water bath at 70 °C. Subsequently, 5 mL of a 50% 2-propanol–water solution was added to each tube 45 min after removal from the water bath, and the absorbance at a wavelength of 510 nm, measured using an OPTIZEN™ POP UV-Visible spectrophotometer (Mecasys Co., Ltd., South Korea), was recorded. The proline content was calculated using the formula (1) below, recommended by the European Commission's Harmonised Methods for Honey, and expressed in mg/kg of honey:

$$Proline \left( \frac{mg}{kg} \right) = \left( \frac{E_S}{E_a} \right) \ x \ \left( \frac{E_1}{E_2} \right) \ x \ 80$$

(1)

where $E_s$ represents the absorbance of the honey sample solution, $E_a$ is the absorbance of the proline standard solution (average of two measurements), $E_1$ denotes the mass of proline used for the stock solution (mg), $E_2$ is the weight of the honey sample (g), and 80 corresponds to the dilution factor.

## 2.9 Honey colour

The instrumental colour measurement was performed using an Envi Sense NH310 colourimeter. The measurement system employed was the CIELAB system, where L* represents brightness as a spatial vector, while $a*$ and $b*$ are trichromatic coordinates. Positive $a*$ values correspond to red, negative values to green; positive $b*$ values correspond to yellow, and negative values to blue. Additionally, the CIE L*, $C_{ab}^*$, and $h_{ab}^{\circ}$ values were presented, where $C_{ab}^*$ denotes chroma and $h_{ab}^{\circ}$ represents the hue angle. These parameters were derived from the $a*$ and $b*$ values on the CIE Lab* scale, with the L* value remaining consistent across scales. The diameter of the measurement aperture was 8 mm. Each test was measured in quadruplicate.

## 2.10 Sugar content

To determine the sugar composition, 1 g of honey was weighed in duplicate into 50 mL graduated flasks. The flasks were filled to volume with distilled water, mixed thoroughly, and filtered. The filtrate was analysed using a Gilson HPLC system equipped with Gilson 306 pumps, a 234 Autoinjector automatic sampler, an amine column (Aminex HPX-87H, BioRad, 300 × 7.8 mm), and a refractive index detector (Knauer K2300). The mobile phase consisted of 0.03 M sulphuric acid (VI), and the separation process was performed at 42°C using a Phenomenex Thermasphere TS130 thermostat, with a flow rate of 0.5 mL/min. Quantitative analysis was conducted using previously determined calibration curves for aqueous solutions of glucose, fructose, maltose, and sucrose, in the range of 0.02 to 20 mg/mL.

## 2.11 Sensory analysis

The sensory analysis was carried out at the Department of Food Quality Analysis and Evaluation, Sensory Analysis Laboratory, University of Life Sciences in Lublin. The sensory panel comprised nine trained assessors certified according to ISO 8586 [21]. The panel evaluated the samples using two complementary methods: (1) a five-point hedonic scale (+2 = "like very much" … –2 = "dislike very much") for taste, aroma, and colour; and (2) a Check-All-That-Apply (CATA) questionnaire including 11 descriptors (five odour descriptors: weak aroma, strong aroma, very strong aroma, herbal, characteristic herbal "individual description"; six taste descriptors: sweet, mild, sharp-scratching, bitter, herbal, characteristic herbal "individual description").



Sample preparation and testing procedure:

Honey samples were prepared following a modified version of the protocol described by Moumeh*et al.* [22]. Algerian honeys were coded as S1–S37 (Table 1), and Polish reference honeys as C38 (multifloral), C39 (multifloral), C40 (heather), and C41 (buckwheat). Each sample (30 g) was presented in a 100 mL transparent jar at 20±2 °C. Every sample was evaluated in three separate sessions. To enhance initial olfactory perception, assessors could apply the honey to the jar walls using a spatula. A bowl of coffee beans was provided to each assessor for nasal palate cleansing. For taste and retronasal aroma assessment, a small amount of honey was placed on the tongue with a disposable spatula and allowed to dissolve for a few seconds without inhalation; subsequent aroma release was performed by exhaling through the nose while the sample remained on the palate. Palate cleansing with water was provided between samples.

This study was conducted in accordance with the Declaration of Helsinki (October 2013, Fortaleza, Brazil), and the protocol was approved by the Ethics Committee of the University of Life Sciences in Lublin (Approval No. UKE/54/2025). Written informed consent was obtained from all panelists; the ethics committee approved the consent procedure. Recruitment period: Panelist recruitment was conducted on 01/07/2025 (start and end), and the sensory evaluation was carried out between 01/07/2025 and 03/07/2025.

### 2.12 Statistical analysis

Statistical analyses were carried out using SPSS software (version 22.0). All experiments were performed in triplicate, and the results were expressed as means±standard deviation (SD). Differences between samples were assessed through one-way ANOVA, followed by Duncan's multiple range test to compare the means at a significance threshold of $p = 0.05$.

Data Hedonic scores for taste, aroma and colour were expressed as means±SD and compared between Algerian and Polish honeys by independent two-sample t-tests ($α = 0.05$). Check-All-That-Apply (CATA) responses were converted into attribute-selection frequencies and analysed by Cochran's Q test to detect overall differences among sample groups; post-hoc pairwise comparisons were performed using McNemar's test with Bonferroni correction. Multivariate analysis of sensory descriptors was conducted by Principal Component Analysis (PCA) on the correlation matrix of standardized CATA-frequency variables and mean hedonic scores (SPSS v.22). To explore sensory-based sample grouping, hierarchical cluster analysis (HCA) was applied to the first two principal-component scores using Ward's method and squared Euclidean distances. All tests were two-tailed, and statistical significance was set at $p < 0.05$.

## 3. Results and discussion

### 3.1 Sample origin

Thirty-seven honey samples (S1–S37) of diverse floral origins were collected from beekeepers in western Algeria (Tlemcen, Ain-Temouchent, Sidi Bel Abbes, Mostaganem, Mascara, Tiaret, Naâma, Bechar), regions marked by contrasting topography, climate, and ecology. Table 2 summarises their physicochemical parameters.

### 3.2 Moisture as an indicator of stability

Moisture is a key determinant of honey maturity, fermentation risk, and crystallisation. Values ranged from 14.67±0.11% to 20.87±0.61%. The highest were thyme (S7: 20.87±0.61%) and multifloral (S17: 20.06±0.92%) honeys from Tlemcen, while the lowest were sage from Naâma (S34: 14.67±0.11%) and Euphorbia from Sidi Bel Abbes (S22: 14.87±0.30%).

Algerian eucalyptus honeys (S25: 17.13±0.11%, S27: 16.13±0.11%) had lower values than Moroccan (19.8±0.1%) [23] and Tunisian honeys (19.12±0.07%) [24]. Carob honeys (S10: 18.40±0.00%, S12: 18.73±0.30%) were also below Moroccan carob (20.0±0.10%) [23]. Conversely, rosemary honeys (S2: 17.13±0.11%, S4: 17.43±0.15%) and multifloral (S3: 18.27±0.23%) were comparable with Tunisian rosemary (17.27±0.01%) [24], Moroccan multifloral (17.8±0.10%) [23], and Turkish multifloral (18.39±1.30%) [25].

**Table 2. Physical and chemical parameters of the analyzed honeys.**

| Sample | Moisture content * (%) | pH * | Free acidity * (meq/kg) | EC * (mS/cm) | HMF * (mg/kg) | Proline * (mg/kg) | $[\alpha]_D^{20}$ * |
|---|---|---|---|---|---|---|---|
| S1 | 17.33 ± 0.30 | 3.61 ± 0.02 | 9.67 ± 1.15 | 0.45 ± 0.01 | 11.27 ± 1.51 | 293.31 ± 0.84 | (-) 12.50 ± 0.25 |
| S2 | 17.13 ± 0.11 | 3.47 ± 0.15 | 12.50 ± 0.87 | 0.26 ± 0.00 | 32.26 ± 0.91 | 431.32 ± 5.32 | (-) 10.89 ± 0.03 |
| S3 | 18.27 ± 0.23 | 3.89 ± 0.03 | 14.00 ± 1.00 | 0.21 ± 0.00 | 6.30 ± 0.65 | 398.73 ± 0.18 | (-) 9.55 ± 0.20 |
| S4 | 17.43 ± 0.15 | 4.85 ± 0.18 | 28.33 ± 1.52 | 0.27 ± 0.01 | 25.70 ± 1.06 | 587.93 ± 0.73 | (-) 9.92 ± 0.06 |
| S5 | 15.46 ± 0.11 | 5.60 ± 0.04 | 13.00 ± 0.87 | 0.25 ± 0.00 | 47.43 ± 2.22 | 265.95 ± 1.28 | (-) 8.85 ± 0.03 |
| S6 | 18.60 ± 0.00 | 4.00 ± 0.03 | 18.67 ± 1.53 | 0.55 ± 0.00 | 33.93 ± 2.94 | 987.08 ± 2.61 | (-) 14.08 ± 0.04 |
| S7 | 20.87 ± 0.61 | 4.29 ± 0.02 | 17.33 ± 0.58 | 0.29 ± 0.00 | 9.73 ± 0.93 | 421.04 ± 4.32 | (-) 10.39 ± 0.25 |
| S8 | 17.07 ± 0.11 | 4.29 ± 0.04 | 11.00 ± 1.00 | 0.47 ± 0.01 | 7.63 ± 1.98 | 725.14 ± 6.43 | (-) 11.32 ± 0.21 |
| S9 | 16.80 ± 0.40 | 4.66 ± 0.03 | 22.67 ± 1.15 | 0.25 ± 0.02 | 3.82 ± 0.59 | 655.21 ± 0.27 | (-) 11.17 ± 0.01 |
| S10 | 18.40 ± 0.00 | 4.39 ± 0.19 | 15.83 ± 1.04 | 0.42 ± 0.01 | 26.55 ± 4.23 | 800.42 ± 2.75 | (-) 8.87 ± 0.02 |
| S11 | 15.93 ± 0.30 | 4.62 ± 0.03 | 30.33 ± 1.53 | 0.44 ± 0.01 | 19.76 ± 0.32 | 924.54 ± 6.14 | (-) 11.02 ± 0.02 |
| S12 | 18.73 ± 0.30 | 4.37 ± 0.02 | 8.00 ± 1.00 | 0.72 ± 0.02 | 40.62 ± 4.41 | 684.42 ± 2.57 | (-) 11.40 ± 0.01 |
| S13 | 15.13 ± 0.11 | 4.11 ± 0.01 | 20.00 ± 1.00 | 0.91 ± 0.01 | 17.81 ± 0.98 | 734.81 ± 2.23 | (-) 10.37 ± 0.09 |
| S14 | 18.07 ± 0.30 | 4.61 ± 0.05 | 11.00 ± 1.00 | 0.51 ± 0.01 | 6.02 ± 0.82 | 359.58 ± 1.76 | (-) 10.59 ± 0.07 |
| S15 | 16.13 ± 0.41 | 4.29 ± 0.02 | 21.33 ± 1.53 | 0.38 ± 0.00 | 8.42 ± 3.74 | 478.75 ± 2.87 | (-) 9.17 ± 0.01 |
| S16 | 18.00 ± 0.00 | 4.27 ± 0.01 | 15.00 ± 1.00 | 0.16 ± 0.01 | 49.43 ± 0.72 | 412.83 ± 4.41 | (-) 8.75 ± 0.02 |
| S17 | 20.06 ± 0.92 | 4.63 ± 0.04 | 19.33 ± 1.52 | 0.42 ± 0.00 | 4.64 ± 0.80 | 871.39 ± 1.06 | (-) 11.66 ± 0.07 |
| S18 | 16.67 ± 0.11 | 3.81 ± 0.09 | 15.67 ± 1.15 | 0.32 ± 0.01 | 35.63 ± 0.42 | 905.82 ± 3.13 | (-) 10.69 ± 0.16 |
| S19 | 19.87 ± 0.41 | 5.10 ± 0.06 | 39.33 ± 1.15 | 1.18 ± 0.02 | 7.91 ± 0.51 | 301.81 ± 0.90 | (-) 10.48 ± 0.02 |
| S20 | 16.33 ± 0.23 | 4.38 ± 0.02 | 10.33 ± 0.11 | 1.04 ± 0.01 | 2.45 ± 0.34 | 1006.34 ± 3.78 | (-) 10.92 ± 0.01 |
| S21 | 18.06 ± 0.23 | 4.19 ± 0.01 | 11.83 ± 0.29 | 0.50 ± 0.00 | 9.48 ± 1.73 | 899.21 ± 2.61 | (-) 9.86 ± 0.02 |
| S22 | 14.87 ± 0.30 | 4.36 ± 0.53 | 15.67 ± 1.15 | 0.24 ± 0.00 | 4.44 ± 0.82 | 458.30 ± 2.19 | (-) 9.51 ± 0.04 |
| S23 | 16.66 ± 0.11 | 4.52 ± 0.43 | 20.33 ± 1.52 | 0.44 ± 0.01 | 7.54 ± 0.64 | 1132.73 ± 2.49 | (-) 10.72 ± 0.02 |
| S24 | 17.60 ± 0.00 | 3.96 ± 0.15 | 20.67 ± 3.05 | 0.16 ± 0.00 | 19.31 ± 0.89 | 794.23 ± 2.37 | (-) 9.70 ± 0.03 |
| S25 | 17.13 ± 0.11 | 4.10 ± 0.01 | 20.33 ± 2.31 | 0.32 ± 0.00 | 18.11 ± 3.17 | 1078.64 ± 8.87 | (-) 10.46 ± 0.20 |
| S26 | 18.07 ± 0.11 | 4.72 ± 0.03 | 25.00 ± 2.00 | 0.32 ± 0.00 | 4.66 ± 0.79 | 491.47 ± 7.21 | (-) 9.46 ± 0.04 |
| S27 | 16.13 ± 0.11 | 4.29 ± 0.02 | 33.67 ± 1.53 | 0.58 ± 0.01 | 10.47 ± 2.35 | 741.25 ± 2.06 | (-) 11.35 ± 0.03 |
| S28 | 17.47 ± 0.23 | 4.65 ± 0.01 | 28.00 ± 2.00 | 0.21 ± 0.00 | 16.77 ± 0.64 | 285.59 ± 0.88 | (-) 10.13 ± 0.02 |
| S29 | 16.80 ± 0.00 | 4.21 ± 0.02 | 40.33 ± 2.52 | 0.25 ± 0.00 | 34.90 ± 2.54 | 1200.66 ± 1.92 | (-) 10.56 ± 0.02 |
| S30 | 15.87 ± 0.30 | 4.01 ± 0.02 | 9.33 ± 0.76 | 0.36 ± 0.00 | 1.79 ± 0.02 | 1117.33 ± 3.63 | (-) 8.30 ± 0.14 |
| S31 | 16.33 ± 0.23 | 3.94 ± 0.01 | 17.16 ± 0.76 | 0.44 ± 0.00 | 4.34 ± 0.26 | 326.50 ± 1.05 | (-) 9.57 ± 0.20 |
| S32 | 15.80 ± 0.40 | 4.69 ± 0.01 | 17.16 ± 1.52 | 0.25 ± 0.00 | 28.29 ± 0.37 | 478.67 ± 0.77 | (-) 10.40 ± 0.03 |
| S33 | 15.27 ± 0.30 | 5.31 ± 0.01 | 12.33 ± 1.52 | 0.22 ± 0.00 | 15.25 ± 3.75 | 637.61 ± 1.35 | (-) 10.10 ± 0.02 |
| S34 | 14.67 ± 0.11 | 4.53 ± 0.07 | 22.67 ± 2.08 | 0.20 ± 0.00 | 36.53 ± 1.27 | 508.88 ± 1.78 | (-) 10.96 ± 0.03 |
| S35 | 16.60 ± 0.20 | 4.99 ± 0.01 | 15.83 ± 0.76 | 0.24 ± 0.00 | 45.36 ± 1.43 | 394.83 ± 0.63 | (-) 10.98 ± 0.02 |
| S36 | 18.20 ± 0.53 | 5.01 ± 0.04 | 9.00 ± 1.00 | 0.46 ± 0.00 | 4.54 ± 0.53 | 279.72 ± 2.74 | (-) 10.59 ± 0.33 |
| S37 | 16.93 ± 0.11 | 4.48 ± 0.06 | 13.83 ± 1.26 | 0.25 ± 0.00 | 40.64 ± 0.63 | 845.61 ± 4.48 | (-) 10.22 ± 0.01 |
| Statistics | | | | | | | |
| Mean | 17.02 | 4.35 | 15.29 | 0.41 | 14.54 | 678.46 | −10.62 |
| SD | 0.26 | 0.08 | 1.03 | 0.00 | 1.12 | 2.97 | 0.10 |

All values are expressed as means of triplicate determinations ± SD. EC: Electrical conductivity; HMF: Hydroxymethyl furfural; $[\alpha]_D^{20}$: Specific optical rotation. *: The results are statistically significant at $p < 0.05$; in the same column, the values are significantly different at $p < 0.001$.



Overall, 12 samples exceeded 18%, a threshold linked to fermentation [26], while two Tlemcen honeys (S7, S17) surpassed the Codex Alimentarius limit of 20%, likely reflecting harvest or handling issues [27]. Conversely, two samples had < 15%, making them overly viscous and prone to crystallisation. Although techniques like pasteurisation or dehydration are sometimes used to adjust honey's moisture, these methods are not always accepted as they can alter the honey's natural state [28].

Regional trends were evident: honeys from drier zones averaged 16.47 ± 0.20%, versus 18.00 ± 0.36% in more humid areas, confirming climatic influence [29]. Botanical origin also played a decisive role [30]. The most humid were from carob (18.56 ± 0.15%; S10, S12), thyme (18.40 ± 0.45%; S7, S11), orange (18.10 ± 0.26%; S16, S28), and camphor (18.07 ± 0.11%; S26). Intermediate levels were recorded for sweet white mustard (17.76 ± 0.05%; S6, S37), lavender (17.33 ± 0.30%; S1), rosemary (16.96 ± 0.05%; S2, S29), milk thistle (16.80 ± 0.11%; S8, S18, S23), eucalyptus (16.63 ± 0.11%; S25, S27), and harmal (16.60 ± 0.20%; S35). The driest were jujube (15.53 ± 0.35%; S32, S33), spurge (14.87 ± 0.30%; S22), and sage (14.67 ± 0.11%; S34).

## 3.3 Acidity profiles and pH balance

Acidity, resulting from more than 30 organic acids of floral and bee origin, strongly influences honey's flavour, colour, and resistance to microbial spoilage. Predominant acids can serve as markers of floral origin, with darker honeys generally displaying higher acidity. pH also differentiates nectar honeys (pH 3.5–4.5) from honeydew honeys (pH 4.5–5.5), regardless of geographical origin.

Measured pH values ranged from 3.47 ± 0.15 (S2) to 5.60 ± 0.04 (S5) (Table 2), fully complying with Codex Alimentarius standards [31]. These results align with previously reported ranges for Algerian honeys (3.75–5.56 [32]; 3.6 ± 0.16–6.2 ± 0.30 [33]; 3.5–4.7 [34]) and are comparable to those documented for honeys from Morocco (4.17 ± 0.05–5.05 ± 0.13 [35]), Tunisia (3.45–4.63 [24]), Palestine (3.03–5.98 [36]), and Spain (4.4 ± 0.20 [37]).

pH variation was clearly linked to floral origin [30]. The most acidic honeys were rosemary (3.47 ± 0.15; S2) and lavender (3.61 ± 0.02; S1), followed by white mustard (4.00 ± 0.03; S6), eucalyptus (4.10 ± 0.01; S25), orange (4.27 ± 0.01; S16), thyme (4.29 ± 0.02; S7), Euphorbia (4.36 ± 0.53; S22), and carob (4.39 ± 0.19; S10). In addition to floral origin, geographical factors, soil type, and mineral content may also contribute to differences in pH [38].

Free acidity ranged from 8.00 ± 1.00 (S12) to 40.33 ± 2.52 meq/kg (S29) (Table 2), all below the Codex Alimentarius maximum of 50 meq/kg [31]. These results align with Algerian honeys (14.1–46.5 meq/kg [39]), Moroccan honeys (11.0 ± 2.29–42.50 ± 2.29 meq/kg [35]), Tunisian honeys (7.11 ± 0.20–27.20 ± 0.20 meq/kg), and Turkish honeys (8.0–46.89 meq/kg) [40]. Variations can be attributed to botanical origin and harvest season [40]. Importantly, none of the samples exceeded regulatory limits, confirming their freshness and high quality.

## 3.4 Electrical conductivity and mineral signature

Electrical conductivity (EC) is a key metric for differentiating between nectar and honeydew honeys. Honeydew honeys typically have a higher mineral content, resulting in a greater EC. Furthermore, darker honeys generally exhibit more efficient electrical conductivity [5,41].

Observed EC values ranged from 0.16 ± 0.01 to 1.18 ± 0.02 mS/cm (Table 2). With the exception of multifloral honeys (S13, S19, S20) that exceeded 0.8 mS/cm, the majority were classified as nectar honeys (EC ≤ 0.8 mS/cm). All samples satisfied the Codex Alimentarius standards [31].

These results agree with previously reported ranges for Algerian honeys (0.29–1.35 mS/cm [39]; 0.133–1.460 mS/cm [34]) and are consistent with values documented for Tunisian honeys (0.39 ± 0.02–0.89 ± 0.06 mS/cm [24]), Turkish honeys (0.19 ± 0.06–1.13 ± 0.25 mS/cm [25]), and Moroccan honeys (0.36 ± 0.02–1.35 ± 0.03 mS/cm [35]). Elevated EC values are generally associated with darker or pollen-rich honeys containing higher concentrations of ionisable minerals, organic acids, and proteins, and may also reflect the presence of both nectar and honeydew components [31].

The floral origin of the honey samples clearly influenced their EC, as reported in earlier studies [30,38,41,42]. The highest values were observed in multifloral honeys (0.91±0.01 mS/cm, S13; 1.18±0.02 mS/cm, S19; 1.04±0.01 mS/cm, S20), followed by carob (0.72±0.02 mS/cm, S12), eucalyptus (0.58±0.01 mS/cm, S27), sweet white mustard (0.55±0.00 mS/cm, S6), milk thistle (0.47±0.01 mS/cm, S8), lavender (0.45±0.01 mS/cm, S1), and thyme (0.44±0.01 mS/cm, S11). In contrast, lower EC values were recorded for camphor (0.32±0.00 mS/cm, S26), rosemary (0.26±0.00 mS/cm, S2), jujube (0.25±0.00 mS/cm, S32), euphorbia (0.24±0.00 mS/cm, S22), harmal (0.24±0.00 mS/cm, S35), orange (0.21±0.00 mS/cm, S28), and sage (0.20±0.00 mS/cm, S34).

Such variability is a robust indicator, directly influenced by the concentration of dissolved components like mineral salts, organic acids, and proteins [43]. The levels of these constituents are, in turn, determined by the honey's botanical origin and the specific regional climatic conditions of its production [44], which dictate its overall chemical composition.

## 3.5 HMF levels and honey freshness

Hydroxymethylfurfural (HMF), a sugar degradation product, is widely recognised as an indicator of honey freshness and heat exposure. Quantification by UV spectrophotometry (White's method) yielded values between 1.79±0.02 and 49.43±0.72 mg/kg (Table 2).

Five samples (S5, S12, S16, S35, and S37) exceeded the Codex Alimentarius limit of 40 mg/kg, recording 47.43±2.22, 40.62±4.41, 49.43±0.72, 45.36±1.43, and 40.64±0.63 mg/kg, respectively. However, all remained below the threshold of 60 mg/kg set for honeys from tropical or warm climates, confirming compliance with international standards. Elevated levels in S12 and S16 may reflect heat exposure or inadequate handling prior to storage, while high values in S5, S35, and S37 likely result from the semi-arid and arid climates of El Aricha, Mecheria, and Bechar. In contrast, 18 samples had HMF levels below 15 mg/kg, demonstrating excellent freshness.

Marked variation was observed by honey type. The lowest values were found in multifloral honeys (1.79±0.02 mg/kg, S30; 2.45±0.34 mg/kg, S20), euphorbia honey (4.44±0.82 mg/kg, S22), and camphor honey (4.66±0.79 mg/kg, S26), while the highest were recorded in orange honey (49.43±0.72 mg/kg, S16) and harmal honey (45.36±1.43 mg/kg, S35). These findings emphasise that HMF levels depend more on processing, storage, and climatic conditions than floral origin.

The results were slightly higher than those previously reported for Algerian honeys (8.80–39.62 mg/kg [45]; 1.30–31.8 mg/kg [39]), Tunisian honeys (12.07±1.00–27.43±1.50 mg/kg [24]), Portuguese honeys (7.4±0.10–28.4±0.10 mg/kg [46]), and Turkish honeys (1.59±1.32–11.83±4.17 mg/kg [25]). Conversely, they were lower than those reported for other Algerian (11.04±0.66–82.00±0.58 mg/kg [47]; 2.84–117.7 mg/kg [48]) and Palestinian honeys (10.16±0.53–81.86±2.64 mg/kg [49]).

Overall, while some samples exhibited elevated HMF, all complied with international standards, confirming their freshness and acceptable quality.

## 3.6 Proline as a marker of maturity

Amino acids constitute about 1.0% of honey, with proline 50–85%. The 26 amino acids vary with origin (nectar or honeydew), and proline declines during storage; it is an indicator of maturity and authenticity. International standards require ≥180 mg/kg. In Table 2, proline ranged from 265.95±1.28 to 1200.66±1.92 mg/kg. Among floral honeys, orange blossom (285.59±0.88 mg/kg, S28) had the lowest level, followed by lavender (293.31±0.84 mg/kg, S1), harmal (394.83±0.63 mg/kg, S35), thyme (421.04±4.32 mg/kg, S7), euphorbia (458.30±2.19 mg/kg, S22), jujube (478.67±0.77 mg/kg, S32), and camphor (491.47±7.21 mg/kg, S26). The highest values were rosemary (1200.66±1.92 mg/kg, S29) and milk thistle (1132.73±2.49 mg/kg). Several exceeded 500 mg/kg, reflecting nectar–pollen ratios. Concentrations matched Moroccan honeys (441.67–1207.55 mg/kg) [50] and Spanish honeys (510±216–1322 mg/kg) [51], and were higher than Algerian (551.88±10.50–852.0±9.90 mg/kg) [47], Palestinian (229.44±3.24–720.87±5.18 mg/kg) [49], Turkish (357.00±34.38–758.56±67.73 mg/kg) [25], and Portuguese honeys

(412.3 ± 4.8–566.6 ± 4.8 mg/kg) [46]. All samples exceeded 180 mg/kg, supporting maturity and purity; variation across and within floral sources reflects botanical origin.

## 3.7 Optical rotation

Honey optical rotation depends on its sugar composition and concentration. In several countries, such as Greece, Italy, and the United Kingdom, it is used to distinguish floral honeys (levorotatory) from honeydew honeys (dextrorotatory). Dextrorotation in honeydew is linked to lower fructose and higher di- and oligosaccharides.

All analysed samples exhibited negative values, including honeydew honeys (Table 2). The most negative was sweet white mustard honey (−14.08 ± 0.04°, S6), followed by lavender honey (−12.50 ± 0.25°, S1). The least negative were multifloral (−8.30 ± 0.14°, S30; −8.85 ± 0.03°, S5), orange (−8.75 ± 0.02°, S16), and carob honeys (−8.87 ± 0.02°, S10). These results agree with Algerian honeys (−14.35 ± 0.03° to −4.65 ± 0.03°) [52], (−18.46 ± 0.76° to −2.07 ± 0.24°) [53], Polish buckwheat (−12.0 ± 1° to −7.5 ± 0.5°) [54], Portuguese honeys (−15.4 ± 0.6° to −11.9 ± 0.8°) [46], and Spanish honeys (−8.94° to −14.13°) [55].

Optical rotation may thus support monofloral discrimination. Its simplicity, speed, and cost-effectiveness warrant consideration for standardised international application.

## 3.8 Sugars and crystallisation potential

Differentiation between pure and adulterated honey is primarily based on sugar composition. Natural honey generally contains ~40% fructose and ~30% glucose, with sucrose usually <5%, except in specific floral varieties [56].

Four sugars—fructose, glucose, sucrose, and maltose—were quantified. Total sugar content ranged from 68.12 ± 0.55% (S35, harmal) to 81.47 ± 0.05% (S20, multifloral) (Table 3). Fructose exceeded glucose in all but two samples, confirming authenticity. The highest fructose concentration was found in multifloral honey (47.47 ± 1.21%, S31), whereas the lowest occurred in harmal honey (33.88 ± 0.34%, S35). Glucose values ranged from 22.07 ± 0.17% (S4, multifloral) to 38.01 ± 0.90% (S37, sweet white mustard). Notably, euphorbia (S22) and sweet white mustard (S37) honeys contained more glucose than fructose, a feature also observed in rapeseed (*Brassica napus*) and dandelion (*Taraxacum officinale*) honeys [57].

Sucrose and maltose were detected in all samples, ranging from 2.24 ± 0.03% (S6) to 8.22 ± 0.02% (S36), averaging 5.41%. Values remained below the 5% limit for sucrose in pure honey, confirming absence of adulteration. Maltose was always <30 mg/g, in agreement with typical natural profiles [58].

Crystallisation tendencies were assessed using the fructose-to-glucose (F/G) and glucose-to-water (G/W) ratios [57,59]. The F/G ratio ranged from 0.90 ± 0.02 (S37) to 1.97 ± 0.02 (S4), with an overall mean of 1.24. Three samples (S37, S22, S5) had ratios <1.0, indicating rapid crystallisation potential [27,56]. Conversely, samples S4 (1.97 ± 0.02) and S31 (1.75 ± 0.00) exhibited fructose predominance and a low glycaemic index, making them beneficial for individuals with impaired glucose tolerance [60].

The G/W ratio, more predictive of crystallization [57,61], exceeded 2.0 in one-third of the samples, consistent with their low moisture levels. None fell below 1.0, indicating stability. Crystallisation, however, is also influenced by insoluble matter and storage temperature [5,62].

Overall, total fructose and glucose contents ranged from 61.71 ± 0.45% to 77.94 ± 0.08%, above the 60 g/100 g threshold of the European Honey Directive [63]. These results are consistent with Algerian (77.72–84.40%) [46], Tunisian (64.71–73.69%) [24], Moroccan (67.06–79.85%) [64], Portuguese (61–78%) [65], and Spanish honeys (63.42–73.43%) [66].

Thus, sugar composition not only confirms the authenticity and quality of Algerian honeys but also highlights variations linked to botanical source, geographical origin, and storage conditions [2].



**Table 3. Sugar content analysis results.**

| Sample | G * (%) | F * (%) | (M+S) * (%) | (F+G) * (%) | F/G * | G/W * | Total sugar content *(%) |
|---|---|---|---|---|---|---|---|
| S1 | 31.48±0.24 | 38.94±0.47 | 5.94±0.22 | 70.42±0.71 | 1.23±0.00 | 1.83±0.04 | 76.37±0.93 |
| S2 | 30.65±1.42 | 40.36±0.40 | 7.11±0.74 | 71.01±1.82 | 1.31±0.05 | 1.79±0.07 | 78.13±2.56 |
| S3 | 32.40±0.20 | 40.76±0.79 | 6.67±0.09 | 73.16±0.99 | 1.25±0.02 | 1.78±0.02 | 79.84±1.08 |
| S4 | 22.07±0.17 | 43.53±0.14 | 7.75±0.22 | 65.60±0.03 | 1.97±0.02 | 1.27±0.01 | 73.35±0.26 |
| S5 | 34.64±0.46 | 34.12±0.15 | 6.39±0.16 | 68.76±0.61 | 0.98±0.00 | 2.25±0.03 | 75.15±0.76 |
| S6 | 32.13±0.00 | 36.22±0.28 | 2.24±0.03 | 68.34±0.28 | 1.12±0.00 | 1.73±0.00 | 70.59±0.25 |
| S7 | 30.01±0.34 | 37.24±0.65 | 8.14±0.01 | 67.26±0.99 | 1.24±0.01 | 1.42±0.00 | 75.40±0.98 |
| S8 | 31.98±0.06 | 41.50±2.06 | 3.76±0.01 | 73.49±2.12 | 1.29±0.06 | 1.87±0.02 | 77.24±2.11 |
| S9 | 34.33±0.08 | 37.88±0.09 | 5.21±0.05 | 72.21±0.18 | 1.10±0.00 | 2.07±0.04 | 77.43±0.23 |
| S10 | 31.10±0.07 | 41.65±0.31 | 5.33±0.05 | 72.76±0.38 | 1.33±0.00 | 1.69±0.01 | 78.09±0.44 |
| S11 | 33.99±2.42 | 40.31±2.07 | 4.54±0.06 | 74.30±4.50 | 1.18±0.02 | 2.11±0.13 | 78.84±4.43 |
| S12 | 34.32±0.09 | 35.87±0.04 | 4.11±0.06 | 70.20±0.05 | 1.04±0.00 | 1.85±0.03 | 74.31±0.11 |
| S13 | 33.07±0.13 | 39.08±0.51 | 3.91±0.07 | 72.15±0.64 | 1.18±0.01 | 2.19±0.03 | 76.06±0.71 |
| S14 | 30.25±0.07 | 41.23±0.48 | 3.87±0.83 | 71.49±0.54 | 1.36±0.01 | 1.66±0.02 | 75.36±1.38 |
| S15 | 28.31±2.63 | 39.91±1.62 | 5.66±0.16 | 68.22±4.26 | 1.41±0.07 | 1.78±0.15 | 73.88±4.42 |
| S16 | 28.26±2.69 | 39.85±1.69 | 5.60±0.23 | 68.11±4.38 | 1.41±0.08 | 1.57±0.15 | 73.71±4.61 |
| S17 | 35.27±0.87 | 38.77±0.68 | 4.22±0.02 | 74.04±1.55 | 1.09±0.00 | 1.71±0.04 | 78.26±1.52 |
| S18 | 34.67±0.35 | 39.93±0.82 | 4.19±0.09 | 74.61±1.17 | 1.15±0.01 | 2.08±0.01 | 78.80±1.26 |
| S19 | 30.81±0.00 | 37.53±0.07 | 6.23±0.01 | 68.35±0.06 | 1.22±0.00 | 1.53±0.01 | 74.58±0.05 |
| S20 | 37.17±0.02 | 40.78±0.09 | 3.53±0.02 | 77.94±0.08 | 1.09±0.00 | 2.29±0.00 | 81.47±0.05 |
| S21 | 31.92±0.61 | 37.34±1.34 | 4.93±0.17 | 69.27±1.95 | 1.17±0.01 | 1.75±0.03 | 74.20±1.78 |
| S22 | 36.99±0.14 | 35.85±0.01 | 5.33±0.09 | 72.84±0.12 | 0.97±0.00 | 2.52±0.02 | 78.18±0.21 |
| S23 | 28.84±0.25 | 39.15±0.27 | 5.11±0.01 | 67.99±0.52 | 1.35±0.00 | 1.73±0.00 | 73.10±0.53 |
| S24 | 27.87±0.06 | 38.67±0.23 | 4.64±0.01 | 66.53±0.29 | 1.38±0.00 | 1.58±0.01 | 71.17±0.28 |
| S25 | 27.87±0.06 | 38.67±0.23 | 4.64±0.01 | 66.53±0.29 | 1.38±0.01 | 1.63±0.01 | 71.17±0.28 |
| S26 | 30.03±0.62 | 43.13±0.97 | 6.08±0.00 | 73.16±0.35 | 1.43±0.06 | 1.66±0.02 | 79.24±0.35 |
| S27 | 30.53±0.48 | 37.45±0.40 | 4.09±0.01 | 67.98±0.88 | 1.22±0.01 | 1.90±0.01 | 72.08±0.90 |
| S28 | 31.16±0.08 | 37.91±0.12 | 7.02±0.06 | 69.07±0.04 | 1.21±0.01 | 1.79±0.03 | 76.09±0.03 |
| S29 | 31.45±0.18 | 41.27±0.11 | 5.90±0.16 | 72.72±0.29 | 1.31±0.00 | 1.87±0.01 | 78.62±0.45 |
| S30 | 35.60±0.01 | 40.13±0.07 | 5.12±0.79 | 75.73±0.08 | 1.13±0.00 | 2.27±0.02 | 80.85±0.88 |
| S31 | 27.16±0.71 | 47.47±1.21 | 6.24±0.09 | 74.63±1.92 | 1.75±0.00 | 1.68±0.04 | 80.87±2.01 |
| S32 | 33.04±0.06 | 39.78±0.25 | 5.88±0.02 | 72.82±0.31 | 1.20±0.01 | 2.07±0.04 | 78.70±0.33 |
| S33 | 30.96±0.04 | 37.53±0.01 | 6.62±0.13 | 68.50±0.04 | 1.21±0.00 | 2.01±0.03 | 75.12±0.17 |
| S34 | 34.50±0.27 | 36.38±0.06 | 5.97±0.18 | 70.89±0.21 | 1.05±0.01 | 2.35±0.01 | 76.86±0.39 |
| S35 | 27.83±0.11 | 33.88±0.34 | 6.41±0.10 | 61.71±0.45 | 1.21±0.01 | 1.67±0.02 | 68.12±0.55 |
| S36 | 30.53±0.03 | 35.39±0.10 | 8.22±0.02 | 65.92±0.07 | 1.16±0.00 | 1.71±0.01 | 74.14±0.05 |
| S37 | 38.01±0.90 | 34.45±0.12 | 3.82±0.16 | 72.47±1.03 | 0.90±0.02 | 2.24±0.05 | 76.29±1.19 |
| Statistics | | | | | | | |
| Mean | 31.35 | 38.52 | 5.57 | 70.66 | 1.22 | 1.85 | 75.89 |
| SD | 0.43 | 0.59 | 0.15 | 0.90 | 0.01 | 0.03 | 1.15 |

All values are expressed as means of triplicate determinations±SD. G: Glucose; F: Fructose; M: Maltose; S: Sucrose; W: Water; *: The results indicate statistical significance at $p < 0.05$.

 

### 3.9 Chromatic diversity of Algerian honeys

Honey colour is a major quality and classification parameter [27], influenced by floral origin, geographical conditions, and pigment content. Instrumental approaches, particularly the CIELAB colour system, provide a standardised method for assessment [67]. This system describes colour using three coordinates—lightness (L*), red/green (a*), and yellow/blue (b*)—from which chroma ($C_{ab}^*$) and hue angle ($h_{ab}^{\circ}$) are derived:

$$C_{ab}^* = \sqrt{a^{*2} + b^{*2}} \tag{2}$$

$$\text{and} \quad h_{ab}^{\circ} = \arctan\left(\frac{b^*}{a^*}\right), \tag{3}$$

The lightness parameter (L*) ranged from 34.18±0.20 to 60.02±2.47 (Table 4), averaging 44.18. Dark amber honeys such as carob (34.18±0.20, S10), eucalyptus (35.12±2.71, S27), and rosemary (35.19±0.14, S29) exhibited the lowest values, while multifloral (60.02±2.47, S13), harmal (57.84±0.45, S35), and thyme honeys (57.30±0.57, S7) recorded the highest, reflecting their brighter appearance.

The a* coordinate revealed strong differences. Carob honey (−0.57±0.07, S10) was the only sample with a negative value, showing a slight green hue. In contrast, orange blossom honey (10.24±0.05, S16) exhibited the most intense red tone, followed by multifloral (10.21±0.05, S15), milk thistle (8.36±1.40, S23), and multifloral (7.39±0.77, S31). The lowest positive values were observed in lighter honeys, including S14, S24, S12, and S22 (0.85–1.34).

The b* values, representing the yellow/blue axis, ranged from 0.78±0.18 to 13.90±0.38 (mean=6.67). Multifloral honey (13.90±0.38, S36) and sage honey (13.84±2.67, S34) were the most yellowish, while darker honeys—euphorbia (2.32±0.28, S22), eucalyptus (3.06±0.07, S27), camphor (3.11±0.04, S26), sweet white mustard (3.37±0.01, S37), rosemary (3.48±0.12, S29), and milk thistle (3.51±0.35, S8)—showed lower values.

Chroma ($C_{ab}^*$), which measures colour intensity, varied from 0.81±0.06 (carob, S10) to 14.21±2.75 (sage, S34), with an average of 7.48. High values were recorded in lighter honeys such as sage (S34), multifloral (S36), milk thistle (S18), harmal (S35), orange blossom (S16), and sweet white mustard (S6), whereas darker honeys, notably carob (S10), euphorbia (S22), and camphor (S26), displayed the lowest intensities.

The hue angle ($h_{ab}^{\circ}$) further differentiated samples: orange blossom (S16) and harmal (S35) exhibited the highest values, while dark amber honeys generally showed low values.

Overall, most samples from western Algeria were classified as dark or dark amber. This observation is consistent with their richness in pigments, phenolics, pollen, and minerals, which characterise honeys from subhumid, semiarid, and arid regions. Colour differences across samples of identical floral origin highlight the influence of microclimate. Standardising protocols remains essential, as methodological variations directly affect CIELAB colour coordinates [68].

### 3.10 Comparative quality of Algerian and Polish honeys

#### 3.10.1 Organoleptic and nutraceutical attributes.
Western Algerian honeys present a wide sensory spectrum and notable nutraceutical potential, reflecting diverse floral sources and distinctive terroir. Colour ranges from light yellow to dark brown, a trait commonly associated with elevated polyphenol concentrations and increased antioxidant activity [69,70]. Aroma profiles extend from delicate floral notes to woody or slightly bitter nuances, linked to volatile compounds such as aldehydes and ketones [71]. Texture varies from creamy to crystallised according to moisture and sugar composition (fructose, glucose), contributing to mouthfeel and overall acceptability [2]. Mineral composition further modulates taste, colour and electrical conductivity [42,72,73]. Collectively, high levels of polyphenols, flavonoids and enzymatic activity impart antioxidant, anti-inflammatory and antimicrobial properties, supporting potential nutraceutical uses for metabolic and cardiovascular health [2,70,71,74].

**Table 4. Spectrophotometric analysis of colour data for western Algerian honey samples.**

| Sample | L* | a* | b* | $C^*_{ab}$ | $h^°_{ab}$ |
|---|---|---|---|---|---|
| S1 | 37.61±0.76 | 2.87±0.24 | 5.31±0.18 | 6.04±0.24 | 61.63±1.90 |
| S2 | 42.80±2.51 | 2.81±0.21 | 4.98±0.21 | 5.72±0.28 | 60.59±0.85 |
| S3 | 40.50±0.40 | 6.79±0.07 | 5.93±0.05 | 9.01±0.08 | 41.15±0.48 |
| S4 | 40.50±1.12 | 6.79±0.42 | 5.93±0.31 | 9.01±0.52 | 41.15±0.27 |
| S5 | 42.74±1.01 | 2.17±0.02 | 7.17±0.81 | 7.49±0.78 | 73.01±1.81 |
| S6 | 42.63±0.58 | 2.23±0.17 | 11.34±0.83 | 11.55±0.84 | 78.85±0.36 |
| S7 | 57.30±0.57 | 1.91±0.07 | 4.81±0.06 | 5.17±0.08 | 68.33±0.53 |
| S8 | 53.69±2.27 | 1.90±0.29 | 3.51±0.35 | 4.01±0.18 | 61.36±5.98 |
| S9 | 37.91±0.68 | 2.89±0.22 | 4.96±0.30 | 5.74±0.37 | 59.79±0.47 |
| S10 | 34.18±0.20 | (−0.57)±0.07 | (−0.57)±0.04 | 0.81±0.06 | 225.09±3.71 |
| S11 | 41.51±3.19 | 1.84±0.37 | 4.49±0.49 | 4.85±0.59 | 67.85±2.06 |
| S12 | 45.06±1.36 | 1.13±0.07 | 5.81±0.24 | 5.92±0.23 | 78.93±0.91 |
| S13 | 60.02±2.47 | 1.99±0.69 | 7.94±0.74 | 8.22±0.55 | 75.65±5.90 |
| S14 | 41.14±4.54 | 0.85±0.23 | 0.78±0.18 | 1.18±0.12 | 43.16±12.35 |
| S15 | 38.55±0.40 | 10.21±0.05 | 7.29±0.37 | 12.55±0.24 | 35.50±1.32 |
| S16 | 38.60±0.40 | 10.24±0.05 | 7.30±0.36 | 12.56±0.17 | 35.45±1.29 |
| S17 | 54.43±6.18 | 2.84±0.35 | 6.33±0.16 | 6.94±0.20 | 65.87±2.75 |
| S18 | 50.86±0.46 | 3.02±0.14 | 13.25±0.86 | 13.59±0.87 | 77.17±0.26 |
| S19 | 42.81±1.76 | 2.97±0.35 | 7.77±0.83 | 8.31±0.90 | 69.10±0.23 |
| S20 | 45.34±0.06 | 2.37±0.04 | 9.34±0.13 | 9.64±0.12 | 75.73±0.45 |
| S21 | 49.45±2.49 | 2.32±0.08 | 5.86±0.44 | 6.30±0.42 | 68.31±1.32 |
| S22 | 48.45±1.78 | 1.34±0.18 | 2.32±0.28 | 2.68±0.33 | 60.04±0.62 |
| S23 | 41.79±2.53 | 8.36±1.40 | 7.56±1.07 | 11.27±1.75 | 42.24±0.87 |
| S24 | 44.41±2.99 | 1.10±0.12 | 5.40±0.40 | 5.52±0.39 | 78.41±1.47 |
| S25 | 37.64±1.13 | 1.86±0.21 | 5.56±0.35 | 5.87±0.40 | 71.51±0.97 |
| S26 | 45.97±2.28 | 2.07±0.09 | 3.11±0.04 | 3.74±0.08 | 56.39±0.79 |
| S27 | 35.12±2.71 | 2.35±0.36 | 3.06±0.07 | 3.86±0.25 | 52.71±4.00 |
| S28 | 41.65±1.10 | 2.10±0.16 | 4.53±0.27 | 4.99±0.31 | 65.10±0.40 |
| S29 | 35.19±0.14 | 2.09±0.04 | 3.48±0.12 | 4.06±0.13 | 59.05±0.44 |
| S30 | 40.14±0.17 | 2.01±0.37 | 6.86±0.12 | 7.15±0.22 | 73.68±2.52 |
| S31 | 37.77±1.70 | 7.39±0.77 | 7.80±0.95 | 10.74±1.22 | 46.52±0.70 |
| S32 | 48.36±3.28 | 2.30±0.16 | 7.28±0.81 | 7.64±0.81 | 72.40±1.18 |
| S33 | 43.10±0.34 | 2.74±0.07 | 8.90±0.32 | 9.32±0.32 | 72.88±0.30 |
| S34 | 49.77±2.60 | 3.21±0.67 | 13.84±2.67 | 14.21±2.75 | 76.98±0.25 |
| S35 | 57.84±0.45 | 1.67±0.11 | 13.06±0.34 | 13.17±0.34 | 82.73±0.38 |
| S36 | 43.60±0.80 | 2.41±0.07 | 13.90±0.38 | 14.11±0.38 | 80.18±0.02 |
| S37 | 46.23±0.35 | 1.73±0.08 | 3.37±0.01 | 3.79±0.04 | 62.89±0.98 |
| Statistics | | | | | |
| Mean | 44.65 | 3.10 | 6.28 | 7.18 | 66.28 |
| SD | 1.68 | 0.28 | 0.50 | 0.51 | 2.13 |

All values are reported as the means of triplicate measurements±standard deviation. L*: clarity (L*=0, black and L*=100, colorless); a*: green/red color component (a*>0, red and a*<0, green); b*: blue/yellow color component (b*>0, yellow and b*<0, blue);$C^*_{ab}$: chroma and $h^°_{ab}$: hue angle. The findings demonstrate statistical significance at $p<0.05$.

### 3.10.2 Physicochemical comparability with Polish varietals.

Comparison of Algerian samples (Tables 2–4) with literature values for Polish honeys (Table 5) indicates substantial overlap. Moisture (14.7–20.9%) is comparable to Polish multiflorals (16.9–20.0%), while pH (3.5–5.6) and electrical conductivity (0.16–1.18 mS/cm) fall within Polish

**Table 5. Literature-based physicochemical parameters of Polish honeys.**

**Physical and chemical parameters**

| Honey type | Moisture content * (%) | pH * | Free acidity * (meq/kg) | EC * (mS/cm) | HMF * (mg/kg) | Proline * (mg/kg) | $[\alpha]_D^{20}$ * |
|---|---|---|---|---|---|---|---|
| multifloral | 17.0 [75] | 3.87 [76] | 30.3 [75] | 0.41 [75] | 6.91-8.42 [77] | 585 [75] | (−11.0)-(−2.2) [78] |
| | 16.9 [76] | 4.1±0.2 [79] | 30 [76] | 0.40 [76] | 0.5-13.9 [78] | 312.1-443.1 [77] | |
| | 18.6 [80] | | 11.9-28.7 [78] | 0.303-0.584 [81] | | | |
| | 18.0-20.0 [81] | | 34.04±25.33 [79] | | | | |
| | 15.7-19.0 [78] | | | | | | |
| heather | 18.3 [75] | 4.07-4.66 [82] | 35.7 [75] | 0.64 [75] | 0.7-14.8 [81] | 861 [75] | (−14.35)–(−15.03) [83] |
| | 18.6-19.9 [81] | 4.25±0.01 [79] | 14.9-33.8 [82] | 0.533-0.583 [81] | | 33.1-92.1 [82] | |
| | 15.4-21.9 [82] | 3.65 [76] | 32.33±1.03 [79] | 0.37-0.82 [82] | | | |
| buckwheat | 19.9 [75] | 3.44-3.80 [54] | 54.7 [75] | 0.43 [75] | 6.4-16.0 [78] | 892 [75] | (−12.7)-(−5.3) [78] |
| | 18.5 [76] | 4.07±0.16 [79] | 45.5 [76] | 0.51 [76] | 3-79 [54] | | (−12.0)-(−7.5) [54] |
| | 16.5 [80] | | 37.8-50.8 [79] | 0.326-0.507 [81] | | | |
| | 18.1-19.9 [81] | | 34.25±10.67 [79] | | | | |
| | 16.5-20.8 [78] | | | | | | |
| | 16.2-20.8 [54] | | | | | | |

**Sugar content.**

| | G * (%) | F * (%) | (M+S) * (%) | (F+G) * (%) | F/G * | Total sugar content * (%) |
|---|---|---|---|---|---|---|
| multifloral | 30.22-35.42 [84] | 33.72-37.70 [84] | 3.50-7.99 [84] | 63.94-71.96 [84] | 1.03-1.13 [84] | 79.5-82.8 [79] |
| | 34.07-37.74 [77] | 41.99-45.24 [77] | 1.12-1.27 [77] | 56.0-84.1 [77] | 1.11-1.32 [77] | |
| | 19.0-36.3 [79] | 37.0-52.0 [79] | | | | |
| heather | 30.27-33.55 [84] | 37.12-40.92 [84] | 4.10-8.42 [84] | 67.39-73.94 [84] | 1.20-1.27 [84] | 71.49-82.36[84] |
| | 25.9-34.3 [82] | 36.5-43.3 [82] | 1.3-3.3 [82] | 62.4-76.1 [82] | 1.12-1.46 [82] | 72.0-72.9 [85] |
| buckwheat | 24.0-31.1 [78] | 39.3-53.8 [78] | | 63.4-80.1 [78] | | 77.8-82.0 [78] |
| | | | | | | 77.6-82.1 [54] |

**Colour data**

| | L* | a* | b* | $C_{ab}^*$ | $h_{ab}^°$ |
|---|---|---|---|---|---|
| multifloral | 57.29 [86] | 5.12 [86] | 34.90 [86] | 22.74±9 [79] | 0.05±0.04 [79] |
| | 40.51 [86] | 3.50 [86] | 29.94 [86] | | |
| | 56.26 [86] | 6.56 [86] | 37.75 [86] | | |
| | 53.7 [80] | 1.7 [80] | 7.2 [80] | | |
| | 42±1.9 [79] | −1.14±1.05 [79] | 23.7±8.99 [79] | | |
| heather | 26±0.4 [79] | 0.54±0.16 [79] | 5.8±0.21 [79] | 5.83±0.2 [79] | 0.09±0.03 [79] |
| buckwheat | 3.38 [86] | 1.89 [86] | 3.86 [86] | 9.29±3.88 [79] | 0.22±0.35 [79] |
| | 8.40 [86] | 8.68 [86] | 9.21 [86] | | |
| | 12.29 [86] | 12.78 [86] | 17.7 [86] | | |
| | 39.1 [80] | 1.8 [80] | −2.6 [80] | | |
| | 33±8.7 [79] | 2.25±3.84 [79] | 8.39±3.48 [79] | | |

EC: Electrical conductivity; HMF: Hydroxymethyl furfural; $[\alpha]_D^{20}$: Specific optical rotation; G: Glucose; F: Fructose; M: Maltose; S: Sucrose; L*: clarity (L*=0, black and L*=100, colorless); a*: green/red color component (a*>0, red and a*<0, green); b*: blue/yellow color component (b*>0, yellow and b*<0, blue);$C_{ab}^*$: chroma and $h_{ab}^°$: hue angle.



nectar-type ranges (≈0.30–0.64 mS/cm). Free acidity (8–40 meq/kg) matches multifloral and heather honeys but remains below typical buckwheat levels (≈55 meq/kg). HMF content averaged ≈15 mg/kg, with only two samples above 40 mg/kg (max ≈ 47 mg/kg), still acceptable under tropical-honey thresholds. Proline concentrations (266–987 mg/kg; mean ≈680 mg/kg) exceed values reported for Polish multiflorals (312–585 mg/kg) yet remain below heather/buckwheat benchmarks (≈860–890 mg/kg), confirming sample maturity. All samples were levorotatory (−14.1 to −8.3°), consistent with reported Polish nectar honeys (−15.0 to −2.2°). Sugar composition (fructose + glucose = 68–81%; F/G ≈ 1.0–1.4) and CIELAB colour metrics (L* 34–60) further indicate authenticity and parity with Polish varietals.

### 3.10.3 Sensory testing and descriptor analysis.

Hedonic testing (Fig 2) showed Algerian honeys to be visually comparable to Polish references (colour: p = 0.459) but lower for taste (+0.18 ± 0.52 vs + 1.22 ± 0.42; p = 0.009) and aroma (−0.26 ± 0.36 vs + 0.72 ± 0.43; p = 0.016). Despite intra-group variability, eight Algerian samples (including rosemary S29; multifloral S14 and S31; eucalyptus S5) exceeded the acceptance threshold (> 0.5), qualifying as premium.

The heat map of mean sensory attributes (Fig 3) provided an overview of perceptual differences between origins, confirming that Algerian samples were generally milder, while Polish honeys displayed stronger sweet and sharp notes.

The CATA outputs further refined these patterns. Mean descriptor frequencies by honey type (Fig 4) revealed that Algerian samples were more frequently associated with "Taste-Mild" and "Herbal," while Polish honeys scored higher for "Taste-Sweet" and "Sharp." The corresponding heat map of descriptor frequencies (Fig 5) highlighted taste as the principal discriminator across samples, whereas colour and aroma intensity were broadly similar between origins.

### 3.10.4 Multivariate analysis and clustering.

Hierarchical clustering of the full CATA profile identified three stable clusters: A — Sweet–Aromatic (all four Polish honeys plus four Algerian samples, e.g., S29, S14, S31, S5), B — Mild/Neutral (majority Algerian), and C — Sharp–Bitter/Herbal (six Algerian samples) (Fig 6). The choice of hierarchical clustering (HCA) is scientifically justified by its demonstrated sensitivity in revealing groupings within complex honey sensory and compositional datasets, as exemplified by its use to distinguish volatile compound profiles in honey samples using HS–SPME–GC–MS [87]. This approach was preferred over non-hierarchical methods because HCA preserves nested relationships and yields dendrogram representations that are intuitive for mapping sensory proximities in descriptor-rich data.

Principal component analysis (PC1 + PC2 ≈ 65% variance; Fig 7) corroborated the cluster structure: PC1 aligned with sweet/strong odour versus sharp/bitter taste, while PC2 related to herbal notes.

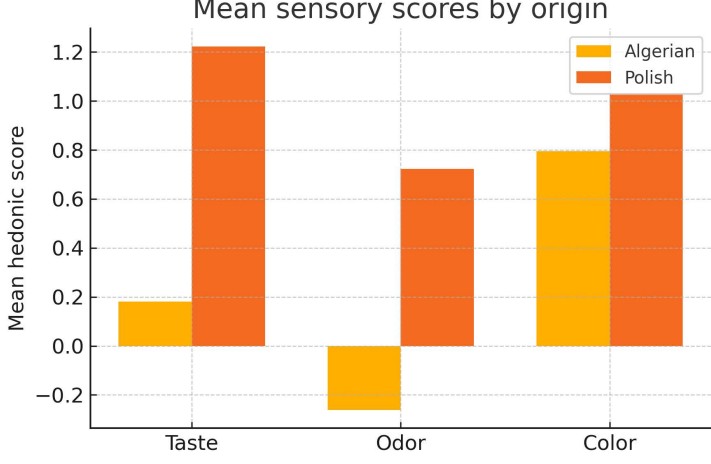

**Fig 2. Mean sensory scores by honey origin.** $p_{Taste}$ = 0.009, pOdor = 0.016, pColor = 0.459.

 

## Heatmap of sensory attribute means

Fig 3. Heat map of the mean sensory attributes for the tested honey samples.

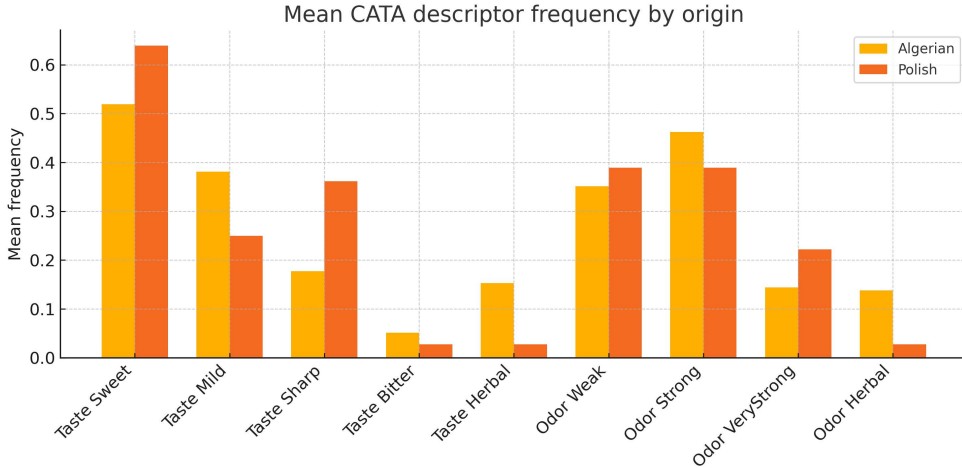

**Fig 4. Mean CATA descriptor frequency by honey origin.**

**3.10.5 Practical implications and market potential.** Descriptor contrasts from CATA (Figs 4 and 5) suggest Algerian honeys are frequently described as "Taste-Mild" and "Herbal," whereas Polish honeys score as "Taste-Sweet" and "Sharp." This implies a market niche: positioning selected Algerian varietals as herbal/functional honeys or employing blending strategies to enhance perceived sweetness. Colour parity with Polish products is an advantage for first-impression marketing; taste and aroma limitations can be addressed through targeted varietal selection (e.g., rosemary S29, milk thistle S18) and refined post-harvest handling (gentle heating, controlled creaming).

Given Poland's import demand (~23,300 t from non-EU sources in 2023) and the limited domestic share of European supply (≈4–6%), there is a tangible commercial opportunity for premium Algerian honeys that meet or exceed quality benchmarks [88,89]. A selective "top-performer" export strategy—promoting honeys that combine high proline, acceptable HMF, and favourable sensory profiles—offers the most viable route to enter premium European niches.

## 4. Conclusions

The integrated physicochemical and sensory evaluation of Western Algerian honeys, in direct comparison with established Polish references, confirmed full compliance with European quality standards. Core compositional indices (moisture, pH, free acidity, HMF, proline, sugar profile, colour) met the required thresholds, while sensory assessment (hedonic scoring, CATA, PCA, hierarchical clustering) revealed three distinct consumer-relevant clusters. Selected samples, notably rosemary (S29) and multifloral honeys (S14, S31), displayed hedonic acceptance and descriptor frequencies comparable to Polish honeys, highlighting their potential for successful positioning in European markets under premium or exotic labels. Certain limitations should nonetheless be acknowledged, particularly the restricted geographic coverage of samples, the moderate number and cultural homogeneity of the sensory panel, and the absence of broader validation across multiple consumer contexts. These factors may constrain the generalisability of the findings. Future research should therefore expand sampling to include additional Algerian regions and harvest seasons, employ larger and more diverse consumer panels within target markets, and integrate longitudinal assessments of product stability. Such developments will strengthen the evidence base for the commercial competitiveness of Algerian honeys in the wider European context.



Fig 5. Heat map of CATA descriptor frequencies across honey samples.

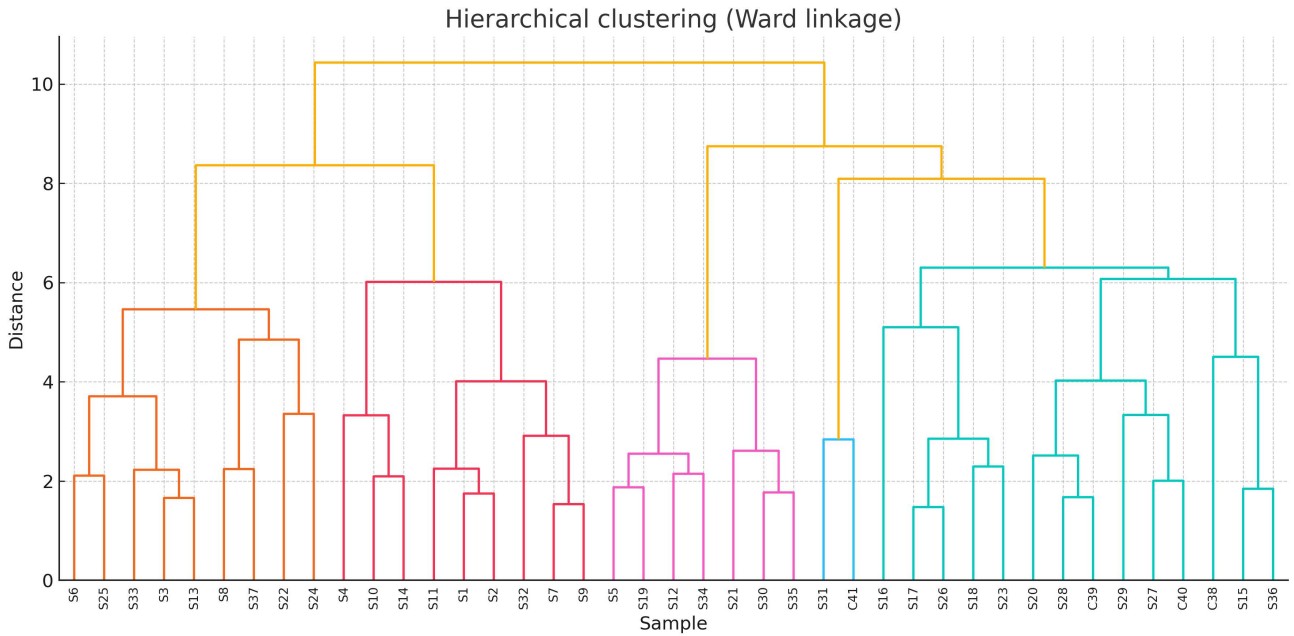

**Fig 6. Hierarchical cluster analysis of full CATA sensory profiles for Algerian and Polish honeys.**

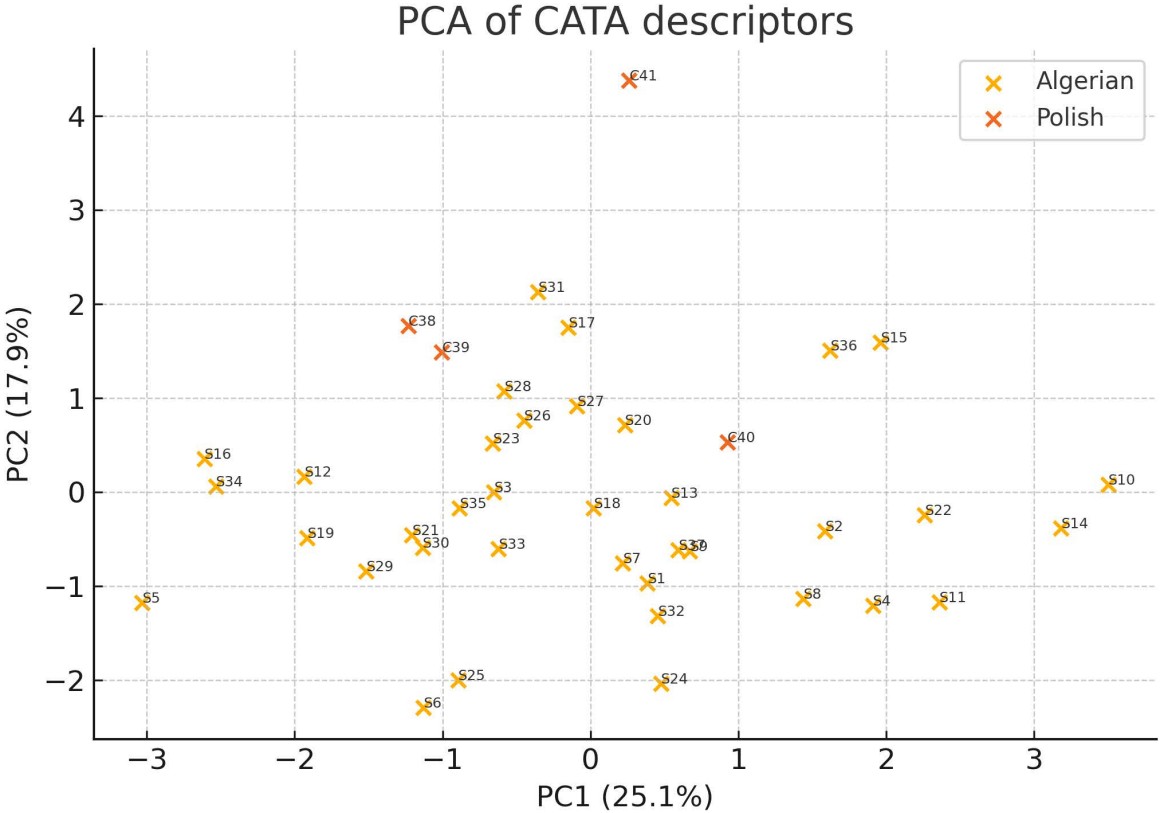

**Fig 7. Principal Component Analysis (PC1 vs. PC2) of combined CATA frequencies and hedonic scores (PC1 + PC2 ≈ 65% of variance).**



## Supporting information

**S1 File. Inclusivity-in-global-research-questionnaire.**
(DOCX)

**S2 File. Source data for the results presented in the article.**
(ZIP)

## Author contributions

**Conceptualization:** Radosław Kowalski, Hocine Allali.

**Data curation:** Dalila Bereksi-Reguig, Grażyna Kowalska, Dariusz Kowalczyk, Jakub Wyrostek, Ewelina Zielińska.

**Formal analysis:** Dalila Bereksi-Reguig, Dariusz Kowalczyk, Jakub Wyrostek, Ewelina Zielińska.

**Funding acquisition:** Grażyna Kowalska.

**Investigation:** Dalila Bereksi-Reguig, Nessrine Kazi Tani, Grażyna Kowalska, Dariusz Kowalczyk, Jakub Wyrostek, Ewelina Zielińska.

**Methodology:** Radosław Kowalski, Hocine Allali.

**Project administration:** Radosław Kowalski, Hocine Allali.

**Resources:** Hocine Allali.

**Software:** Salim Bouchentouf.

**Supervision:** Radosław Kowalski, Hocine Allali.

**Validation:** Dalila Bereksi-Reguig, Salim Bouchentouf, Nessrine Kazi Tani.

**Visualization:** Radosław Kowalski, Hocine Allali.

**Writing – original draft:** Hocine Allali.

**Writing – review & editing:** Radosław Kowalski, Hocine Allali.

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
