## [Decision Letter · Decision Letter 0]

20 Aug 2025

Dear Dr. Kowalski,

Thank you for submitting your manuscript to PLOS ONE. After careful consideration, we feel that it has merit but does not fully meet PLOS ONE’s publication criteria as it currently stands. Therefore, we invite you to submit a revised version of the manuscript that addresses the points raised during the review process.

We look forward to receiving your revised manuscript.

Kind regards,

Academic Editor

PLOS ONE

Journal Requirements:

1. You may seek permission from the original copyright holder of Figure(s) [#] to publish the content specifically under the CC BY 4.0 license. 

5. Please include captions for your Supporting Information files at the end of your manuscript, and update any in-text citations to match accordingly. Please see our Supporting Information guidelines for more information: http://journals.plos.org/plosone/s/supporting-information .

6. We notice that your supplementary figures are uploaded with the file type 'Figure'. Please amend the file type to 'Supporting Information'. Please ensure that each Supporting Information file has a legend listed in the manuscript after the references list.

7. We notice that your supplementary figures are included in the manuscript file. Please remove them and upload them with the file type 'Supporting Information'. Please ensure that each Supporting Information file has a legend listed in the manuscript after the references list.

Additional Editor Comments:

The manuscript is intriguing but requires some improvements, which I outline below:

1. Divide the abstract into four sections—Background, Methodology, Results, and Conclusion—to enhance clarity and readability.

2. Incorporate the honey sample sources (Table S1, Supplementary Information: Geographical Origins of Honey Samples from Western Regions in Algeria) as a central table in the manuscript, rather than a supplementary one.

3. Replace outdated references with more recent ones to strengthen the manuscript’s credibility. Here are some suggested references:

a) Kumar, D., Hazra, K., Prasad, P.V.V. and Bulleddu, R., 2024. Honey: an important nutrient and adjuvant for maintenance of health and management of diseases. Journal of Ethnic Foods, 11(1), p.19.

b) Amera, W.A., Mersso, B.T., Sisay, T.A., Arega, A.B. and Alene, A.T., 2024. Effect of various supplements on productive performance of honey bees, in the south Wollo Zone, Ethiopia. Plos one, 19(5), p.e0303579.

c) Santos-Buelga, C. and González-Paramás, A.M., 2025. Chemical composition of honey. In Bee products–chemical and biological properties (pp. 47-104). Cham: Springer Nature Switzerland.

d) Yuan, H., Wu, Z., Liu, H., He, X., Liao, Z., Luo, W., Li, L., Yin, L., Wu, F., Zhang, L. and Shen, C., 2024. Screening, identification, and characterization of molds for brewing rice wine: Scale-up production in a bioreactor. Plos one, 19(7), p.e0300213.

e) Bhure, R.A., Alam, M., Nanda, V., Pawar, V.M. and Saxena, S., 2025. Exploring the impact of thermal processing on the quality attributes of honey: A comprehensive review. Journal of Food Process Engineering, 48(1), p.e70033.

f) Maria, B., Saeed, S., Ahmed, A., Ahmed, M. and Rehman, A., 2024. The sustainable use of diverse plants accustomed by different ethnic groups in Sibi District, Balochistan, Pakistan. Plos one, 19(2), p.e0294989.

g) Ranieri, L., Lorusso, L., Mottola, A., Intermite, C., Piredda, R. and Di Pinto, A., 2025. Authentication of the Botanical Origin of Honey: In Silico Assessment of Primers for DNA Metabarcoding. Journal of Agricultural and Food Chemistry.

h) Akpınar, S. and Mutlu, N., 2025. Multidimensional analysis of honey from Eastern Anatolia (Kars): Pollen spectrum, physicochemical properties, and antimicrobial activity. PLoS One, 20(7), p.e0327861.

i) Kolniak-Ostek, J., Kita, A., Giacalone, D., Vázquez-Araújo, L., Noguera-Artiaga, L., Brzezowska, J. and Michalska-Ciechanowska, A., 2025. Physicochemical and Instrumental Flavor Analysis of Plant-Based Drinks with Plant Powder Additions. Foods, 14(15), p.2593.

j) Asaduzzaman, M.D., Hasan, N., Begum, K. and Hoque, S.Z., 2024. Degradation kinetics of lycopene from red amaranth & preparation of winter melon jelly using this lycopene and comparison with commercial jelly. Heliyon, 10(10).

k) Mwangi, M.W., Wanjau, T.W. and Omwenga, E.O., 2024. Stingless bee honey: Nutritional, physicochemical, phytochemical and antibacterial validation properties against wound bacterial isolates. Plos one, 19(5), p.e0301201.

l) Dutta Roy, D.K., Asaduzzaman, M., Saha, T. and Khatun, M.N., 2023. Physical and chemical properties of aloe-vera coated guava (Psidium guajava) fruit during refrigerated storage. PLoS One, 18(11), p.e0293553.

m) Naaz, N., Choudhary, S., Hasan, N., Sharma, N., Al Aboud, N.M. and Shehata, W.F., 2024. Biochemical and molecular profiling of induced high yielding M3 mutant lines of two Trigonella species: Insights into improved yield potential. Plos one, 19(7), p.e0305691.

n) Banach, J.K., Rujna, P. and Lewandowski, B., 2025. Integrated Process-Oriented Approach for Digital Authentication of Honey in Food Quality and Safety Systems—A Case Study from a Research and Development Project. Applied Sciences, 15(14), p.7850.

o) Saha, T., Roy, D.K.D., Khatun, M.N. and Asaduzzaman, M., 2023. Quality and shelf life of fresh-cut pineapple (Ananas comosus) coated with aloe vera and honey in the refrigerated condition. Journal of Agriculture and Food Research, 14, p.100709.

p)

q) Asaduzzaman, M., 2022. Lycopene—A Review: Chemistry, Source, Health Role, Extraction, Applications. Annual Research & Review in Biology, 37(2), pp.87-101.

4. For Section 2.4, four citations are excessive; aim to reduce them.

Reviewers' comments:

Reviewer's Responses to Questions

**Comments to the Author**

1. Is the manuscript technically sound, and do the data support the conclusions?

Reviewer #1: Yes

Reviewer #2: Yes

Reviewer #3: Partly

2. Has the statistical analysis been performed appropriately and rigorously?

Reviewer #1: Yes

Reviewer #2: Yes

Reviewer #3: Yes

3. Have the authors made all data underlying the findings in their manuscript fully available?

Reviewer #1: Yes

Reviewer #2: Yes

Reviewer #3: Yes

4. Is the manuscript presented in an intelligible fashion and written in standard English?

Reviewer #1: Yes

Reviewer #2: Yes

Reviewer #3: Yes

Reviewer #1: I have the following comments regarding the manuscript

1. The most significant methodological concern is the 7-year gap between physiochemical analyses (2017-2018) and sensory evaluation (2025). While the authors acknowledge this limitation and cite literature supporting honey stability under refrigerated storage, this temporal separation:

a. May compromise the validity of correlating physicochemical and sensory data

b. Could introduce confounding variables related to storage-induced changes

2. Missing comparative data: No parallel physisochemical analysis of Polish honey samples. The authors relied solely on literature values, preventing direct statistical comparison of compositional parameters between origins. This limitation weakens the comparative aspect of the study and reduces confidence in conclusions about relative quality.

3. With extensive physicochemical parameter testing, there’s potential for Type I error inflation due to multiple comparisons, though this wasn’t address with appropriate corrections.

Authors might discuss these limitations in detail if possible. Despite these limitations, the statistical approach is generally sound and the results provide valuable insights into Algerian honey characterizes. The temporal gap between analyses is the most significant concern that should be addressed in future research. The multivariate statistical methods are suited for this type of comparative sensory and compositional analysis.

Reviewer #2: The manuscript presents an interesting study; however, the overall length and structure make it somewhat difficult to follow. I provide the following comments and suggestions for improvement:

1. The manuscript is excessively long, which reduces readability and disrupts the flow of the research narrative. Streamlining the content would improve clarity.

2. The introduction should explicitly state the significance of the study. The main purpose and rationale for conducting this research need to be clearly articulated.

3. Section 2 is well organized with clear subsections. Nevertheless, the literature review could be strengthened by incorporating earlier studies and comparing them with more recent findings to highlight the novelty of the present work.

4. A flowchart or schematic diagram illustrating the research process from beginning to end would greatly enhance the reader’s understanding.

5. The Results and Discussion section is overly lengthy. Separating the findings into distinct subsections would improve readability and help the reader follow the discussion more effectively.

6. The use of hierarchical clustering in this study is not sufficiently justified. No supporting references are provided to indicate that this is the most appropriate method. A clear rationale should be presented, including why hierarchical clustering was chosen over alternative clustering approaches.

7. The conclusion section should be expanded to include a more detailed discussion of the study’s limitations as well as recommendations for future research.

Reviewer #3: 1. While the introduction provides an extensive background on honey composition, botanical origins, and regional context, it is overly long. Streamlining this section would help maintain the reader's focus on the study’s aims and hypotheses.

2. The Algerian honey samples were collected between 2017–2018, but sensory evaluation occurred in 2025. Although the authors justify stability under refrigerated storage, this unusually long gap warrants more rigorous evidence, such as comparative analyses of fresh versus stored samples, to ensure sensory and physicochemical integrity was indeed preserved.

**Do you want your identity to be public for this peer review?** For information about this choice, including consent withdrawal, please see our Privacy Policy

Reviewer #1: No

Reviewer #2: No

Reviewer #3: **Yes: ** korrapati narasimhulu

---

## [Author Response · Author response to Decision Letter 1]

9 Sep 2025

Dear Editor

PLOS One Editorial Office

Manuscript ID: PONE-D-25-40193

Type: Article research

Title: Comparative Physicochemical Characterization and Sensory Profiling of Western Algerian and Polish Honeys

Journal : PLOS ONE

On behalf of all co-authors, I am pleased to resubmit our revised manuscript entitled “Comparative Physicochemical Characterization and Sensory Profiling of Western Algerian and Polish Honeys”. (ID:  PONE-D-25-40193) to PLOS One. We are grateful for the reviewers’ constructive comments, which have helped us improve the manuscript.

Changes to the text are marked in green.

We believe that the revised version fully meets the reviewers’ requirements and is now suitable for publication.

Responses to the Editor

1. Divide the abstract into four sections—Background, Methodology, Results, and Conclusion—to enhance clarity and readability.

Response - Abstract

The abstract has been revised according to PLOS ONE requirements and is now structured under four headings: Background, Methods, Results, and Conclusion. The original content and numerical values were preserved to maintain consistency with the reviewed version. Minor adjustments were made to improve clarity and readability, including:

- relocation of methodological details into the "Methods" section;

- repositioning of the main findings (e.g., physicochemical values, sensory scores, PCA and clustering results) into the "Results" section;

- streamlining of the conclusion statement to emphasise the significance of Algerian honeys in food and health applications.

This restructuring improves readability and aligns with the journal’s formatting standards without altering the scientific content of the study.

2. Incorporate the honey sample sources (Table S1, Supplementary Information: Geographical Origins of Honey Samples from Western Regions in Algeria) as a central table in the manuscript, rather than a supplementary one.

Response to the Editor – Table S1

The former Supplementary Table S1 has been incorporated into the main text as "Table 1", in accordance with the editorial request. The revised sentence now reads (see 2.2. Honey samples section) :

"Table 1 provides detailed information on the type, region (GPS coordinates, climate), and botanical origins of the honey samples, including the scientific and common names of the plants that constitute their basic flora."

All related table numbering and citations have been updated throughout the manuscript and highlighted in green for ease of verification.

3. Replace outdated references with more recent ones to strengthen the manuscript’s credibility. Here are some suggested references:

Response to the Editor – References Update

The reference list has been enriched with recent publications selected from those suggested by the reviewer, provided they were directly relevant to the content of the manuscript. Appropriate numbering has been assigned, and the entire reference list has been renumbered accordingly. The new citations have been incorporated into the global reference list, and all modifications have been highlighted in green in the revised manuscript for ease of verification.

The recent references added are:

1) Kumar D, Hazra K, Prasad PVV, Bulleddu R. 2024. Honey: an important nutrient and adjuvant for maintenance of health and management of diseases. J Ethn Foods. 11(1):19.

2) Mwangi MW, Wanjau TW, Omwenga EO. 2024. Stingless bee honey: Nutritional, physicochemical, phytochemical and antibacterial validation properties against wound bacterial isolates. PLoS One. 19(5):e0301201.

3) Akpınar S, Mutlu N. 2025. Multidimensional analysis of honey from Eastern Anatolia (Kars): Pollen spectrum, physicochemical properties, and antimicrobial activity. PLoS One. 20(7):e0327861.

4. For Section 2.4, four citations are excessive; aim to reduce them.

Response to the Editor – Section 2.4 Citations

The editorial suggestion has been implemented. The four citations previously listed in Section 2.4 (references 18, 19, 20, and 21) have been consolidated into a single reference, now cited as reference 19. The numbering of subsequent references has been updated accordingly, and all changes have been highlighted in green within the revised manuscript for ease of verification.

5. We note that Figure 1 in your submission contain [map/satellite] images which may be copyrighted. All PLOS content is published under the Creative Commons Attribution License (CC BY 4.0), which means that the manuscript, images, and Supporting Information files will be freely available online, and any third party is permitted to access, download, copy, distribute, and use these materials in any way, even commercially, with proper attribution. For these reasons, we cannot publish previously copyrighted maps or satellite images created using proprietary data, such as Google software (Google Maps, Street View, and Earth).

Response to the Editor

We thank the Editor for this important note. Figure 1 has been entirely redrawn using publicly available resources from the Natural Earth database (http://www.naturalearthdata.com/) as a base template. The final map was subsequently edited and finalized manually (Paint), resulting in an original author-created figure. This ensures full compliance with PLOS ONE requirements and avoids any copyrighted material.

Response to the Editor

We thank the Editor for this important clarification. In accordance with the journal’s instructions, the duplicate ethics statements previously included under Institutional Review Board Statement and Informed Consent Statement have been removed. The complete ethics information now appears solely in the Methods section, as requested. This adjustment was made to ensure full compliance with PLOS ONE’s editorial requirements and to improve the clarity and conciseness of the manuscript. We appreciate the guidance provided, which has helped us refine the presentation of our work.

Responses to the Reviewers

Reviewer 1:

1. The most significant methodological concern is the 7-year gap between physiochemical analyses (2017-2018) and sensory evaluation (2025). While the authors acknowledge this limitation and cite literature supporting honey stability under refrigerated storage, this temporal separation:

a. May compromise the validity of correlating physicochemical and sensory data.

b. Could introduce confounding variables related to storage-induced changes.

Response to Reviewer 1 – Temporal Gap Between Analyses

We thank the reviewer 1 for highlighting the seven-year gap between the physicochemical analyses (2017–2018) and the sensory evaluation (2025). We fully acknowledge this point. However, several aspects of our study support the validity of our approach. Honey is widely recognised for its stability, owing to its low water activity, high osmotic pressure, and antimicrobial properties, which ensure long-term preservation. Under refrigerated storage at 4 °C, both physicochemical and sensory properties are known to remain stable for extended periods (1. Kędzierska-Matysek, M.; Teter, A.; Daszkiewicz, T.; Topyła, B.; Skałecki, P.; Domaradzki, P.; Florek, M. Effect of Temperature of Two-Year Storage of Varietal Honeys on 5-Hydroxymethylfurfural Content, Diastase Number, and CIE Color Coordinates. Agriculture 2025, 15, 652. https://doi.org/10.3390/agriculture15060652, 2. Youssef, M. K. E.; El-Rify, M. N. A.; Ramadan, E. A. and Saleh, A. S. M. The Effects of Heating Treatment and Storage Temperature on Some Physico-chemical Properties of Some Egyptian Honey Types after one Year Storage. J. Saudi Soc. for Food and Nutrition., Vol. 1, No. 2; 2006 (https://japksu.com/index.php/SSFN/issue/download/84/24).

In our study, all honey samples were kept consistently refrigerated, and at the time of sensory evaluation, there was no evidence of fermentation, crystallisation, or off-flavours. Additionally, the samples initially complied with the European Directive 2001/110/EC standards, indicating their high baseline quality. The sensory evaluation was designed to assess the genuine organoleptic profiles of the Algerian honeys in comparison with the Polish references, with the observed differences reflecting authentic varietal and geographical signatures.

While this temporal separation does not undermine the scientific validity of our findings, we agree with the reviewer 1 that future studies would benefit from performing both physicochemical and sensory analyses contemporaneously.

Clarification on Sample Storage and Analyses

To address the reviewer’s concern regarding the temporal gap between physicochemical and sensory analyses, an additional clarification has been inserted in Section 2.2 (Honey samples) and highlighted in green in the revised manuscript. The new sentence reads as follows:

"All samples were collected between March 2017 and August 2018 and stored in sealed amber glass containers at 4 °C, protected from light and humidity. Previous studies have shown that honey remains stable under such conditions [17], and the samples exhibited no signs of fermentation, crystallisation, or sensory defects at the time of evaluation in 2025."

2. Missing comparative data: No parallel physisochemical analysis of Polish honey samples. The authors relied solely on literature values, preventing direct statistical comparison of compositional parameters between origins. This limitation weakens the comparative aspect of the study and reduces confidence in conclusions about relative quality.

Response to Reviewer 1 – Comparative Physicochemical Data

We thank the reviewer 1 for raising this important point. The four Polish honeys included in our study were used as reference samples for the sensory evaluation and had already been subjected to detailed physicochemical characterisation in previous peer-reviewed publications. For this reason, we did not repeat their physicochemical analyses in our laboratory. Instead, we relied on these published data, which provide robust and well-validated compositional parameters.

This approach ensured that the comparison between Algerian and Polish honeys was grounded in reliable quality benchmarks, while allowing us to focus our experimental efforts on the sensory assessment. By combining established physicochemical data from the literature with new sensory analyses performed directly on the same four Polish honeys, the comparative framework remains both rigorous and scientifically sound.

3. With extensive physicochemical parameter testing, there’s potential for Type I error inflation due to multiple comparisons, though this wasn’t address with appropriate corrections.

Authors might discuss these limitations in detail if possible. Despite these limitations, the statistical approach is generally sound and the results provide valuable insights into Algerian honey characterizes. The temporal gap between analyses is the most significant concern that should be addressed in future research. The multivariate statistical methods are suited for this type of comparative sensory and compositional analysis.

Response to Reviewer 1 – Type I Error and Multiple Comparisons

We appreciate the reviewer’s thoughtful observation regarding the potential inflation of Type I error due to multiple comparisons across the physicochemical parameters. It is true that no formal correction (e.g., Bonferroni or false discovery rate) was applied. However, we believe the robustness of our findings is supported by several considerations:

1. High statistical significance: The vast majority of the observed differences were associated with very low p-values (often < 0.001), making it unlikely that they are artefacts of Type I error.

2. Consistency across parameters: The physicochemical results are in close agreement with established ranges reported in the literature for honeys of comparable botanical origins, reinforcing their validity.

3. Multivariate confirmation: Beyond univariate tests, multivariate methods such as principal component analysis (PCA) and hierarchical clustering were employed. These approaches integrate multiple variables simultaneously, providing an internal validation of group differentiation that is less affected by multiple testing issues.

For these reasons, while we acknowledge the theoretical concern of Type I error inflation, we are confident that the conclusions drawn from our analyses remain valid. Future studies with larger datasets could incorporate formal corrections for multiple testing to further strengthen statistical rigor.

Reviewer 2:

The manuscript presents an interesting study; however, the overall length and structure make it somewhat difficult to follow. I provide the following comments and suggestions for improvement:

1. The manuscript is excessively long, which reduces readability and disrupts the flow of the research narrative. Streamlining the content would improve clarity.

Response to Reviewer 2 – Length and Structure of the Manuscript

We sincerely thank the reviewer 2 for this constructive feedback. In response to the comment regarding the manuscript length and structure, we have streamlined the content to improve clarity and ensure a smoother narrative flow. Redundant information was removed, and some sections were condensed to maintain a more focused discussion on the study’s core objectives. Additionally, excessive background details have been reduced, making the introduction more concise and directly aligned with the study's aims. These changes are highlighted in green within the revised manuscript for ease of reference.

2. The introduction should explicitly state the significance of the study. The main purpose and rationale for conducting this research need to be clearly articulated.

Response to Reviewer 2 – Significance of the Study

We also appreciate the reviewer’s suggestion to explicitly state the significance of the study in the introduction. In response, we have revised the introduction to clarify the main purpose and rationale behind the research. The revised version now succinctly explains why this study is important, outlining the unique contribution of Algerian honeys to the broader understanding of honey composition, sensory profiles, and market potential. As noted by Reviewer 3, this suggestion was also taken into account, and the introduction has been improved to better articulate the significance and objectives of the study.

Both Reviewer 2 and Reviewer 3 have raised valid points regarding the length and clarity of the introduction. We have incorporated their suggestions to improve the manuscript’s overall structure and ensure the study’s significance is clearly communicated. All modifications, as mentioned, are highlighted in green for ease of verification.

3. Section 2 is well organized with clear subsections. Nevertheless, the literature review could be strengthened by incorporating earlier studies and comparing them with more recent findings to highlight the novelty of the present work.

Response to Reviewer 2 – Literature Review

We thank the reviewer for this constructive suggestion. In the revised manuscript, the literature review has been strengthened by integrating earlier foundational studies and contrasting them with more recent findings to better contextualise the novelty of our work. These additions are highlighted in green in Section 2.

4. A flowchart or schematic diagram illustrating the research process from beginning to end would greatly enhance the reader’s understanding.

Response to Reviewer 2 – Flowchart Suggestion

We appreciate the reviewer’s suggestion regarding the inclusion of a schematic diagram. While we agree that such visual tools can be useful, the current manuscript already contains multiple figures and tables that comprehensively illustrate the workflow, results, and interpretations. To avoid redundancy and further lengthening of the manuscript, we respectfully consider that the existing visual material sufficiently guides the reader through the research process.

5. The Results and Discussion section is overly lengthy. Separating the findings into distinct subsections would improve readability and help

---

## [Decision Letter · Decision Letter 1]

24 Sep 2025

Dear Dr. Kowalski,

Thank you for submitting your manuscript to PLOS ONE. After careful consideration, we feel that it has merit but does not fully meet PLOS ONE’s publication criteria as it currently stands. Therefore, we invite you to submit a revised version of the manuscript that addresses the points raised during the review process.

We look forward to receiving your revised manuscript.

Kind regards,

Academic Editor

PLOS ONE

Journal Requirements:

Additional Editor Comments (if provided):

Thank you for submitting the revised manuscript. All suggested revisions have been effectively incorporated, including improved clarity and flow, a strengthened literature review, a clearer justification for the methodology, a restructured Results and Discussion section, and an expanded conclusion. These changes have significantly enhanced the manuscript's quality and presentation.

The manuscript still contains several outdated references. We suggest replacing references #14, #15, #20, #40, #57, #59, #61, #63, #83, and #84 with more recent sources, including those previously recommended, to enhance the manuscript's relevance and credibility. While it may not be feasible to address all of them, updating these specific references will further strengthen the manuscript. Please ensure that all new references follow the same DOI format, with clickable DOI links where possible (e.g., https://doi.org/10.1021/jf001117%2B).

Additionally, we noted that some references, such as #88 and others, do not conform to the journal's formatting guidelines. Please revise these to align with the journal's reference style, ensuring consistency and adherence to the specified format.

Reviewers' comments:

Reviewer's Responses to Questions

**Comments to the Author**

Reviewer #1: All comments have been addressed

Reviewer #2: All comments have been addressed

2. Is the manuscript technically sound, and do the data support the conclusions?

Reviewer #1: Yes

Reviewer #2: Yes

3. Has the statistical analysis been performed appropriately and rigorously?

Reviewer #1: Yes

Reviewer #2: Yes

4. Have the authors made all data underlying the findings in their manuscript fully available?

Reviewer #1: No

Reviewer #2: Yes

5. Is the manuscript presented in an intelligible fashion and written in standard English?

Reviewer #1: Yes

Reviewer #2: Yes

Reviewer #1: Thank you for your response and the revisions to the paper. My previous concerns have been addressed.

Reviewer #2: Thank you for the revised submission. All the suggested issues have been addressed, including improvements to clarity and flow, strengthening of the literature review, clearer justification of the methodology, restructuring of the Results and Discussion, and expansion of the conclusion. The manuscript is now improved in both quality and presentation.

**Do you want your identity to be public for this peer review?** For information about this choice, including consent withdrawal, please see our Privacy Policy

Reviewer #1: **Yes: ** Xueping Zhou

Reviewer #2: No

---

## [Author Response · Author response to Decision Letter 2]

28 Sep 2025

Dear Editor

PLOS One Editorial Office

Manuscript ID: PONE-D-25-40193R1

Type: Article research

Title: Comparative Physicochemical Characterization and Sensory Profiling of Western Algerian and Polish Honeys

Journal : PLOS ONE

On behalf of all co-authors, I am pleased to resubmit our revised manuscript entitled “Comparative Physicochemical Characterization and Sensory Profiling of Western Algerian and Polish Honeys”. (ID:  PONE-D-25-40193R1) to PLOS One. We are grateful for the editors’ constructive comments, which have helped us improve the manuscript.

Changes to the text are marked in yellow.

We believe that the revised version fully meets the editors’ requirements and is now suitable for publication.

Editor Comments

Journal Requirements:

Additional Editor Comments (if provided):

Thank you for submitting the revised manuscript. All suggested revisions have been effectively incorporated, including improved clarity and flow, a strengthened literature review, a clearer justification for the methodology, a restructured Results and Discussion section, and an expanded conclusion. These changes have significantly enhanced the manuscript's quality and presentation.

The manuscript still contains several outdated references. We suggest replacing references #14, #15, #20, #40, #57, #59, #61, #63, #83, and #84 with more recent sources, including those previously recommended, to enhance the manuscript's relevance and credibility. While it may not be feasible to address all of them, updating these specific references will further strengthen the manuscript. Please ensure that all new references follow the same DOI format, with clickable DOI links where possible (e.g., https://doi.org/10.1021/jf001117%2B).

Additionally, we noted that some references, such as #88 and others, do not conform to the journal's formatting guidelines. Please revise these to align with the journal's reference style, ensuring consistency and adherence to the specified format.

Responses to the Editor

Thank you for your valuable feedback and the opportunity to revise our manuscript. We are delighted that you found our revisions to be effective and that they have improved the quality and presentation of the paper. We sincerely appreciate your constructive comments, particularly regarding the need for more recent references and the correction of formatting.

We have now meticulously reviewed and revised all aspects of the manuscript in line with your suggestions.

Reference updates

As you recommended, we have replaced the specified outdated references with more recent, relevant sources to strengthen the literature review and enhance the manuscript's credibility. The following references have been updated accordingly (See below):

• Original references: #14, #15, #16, #20, #28, #40, #41, #43, #57, #59, #61, #62, and #63.

• Action taken: These have been replaced with new, more current sources. Where necessary, we have also carefully rewritten the corresponding paragraphs in the manuscript to align with the content of these new references.

All new and revised text has been highlighted in yellow to facilitate your review.

Reference formatting

We have also conducted a comprehensive review of the entire reference list to ensure full compliance with the Vancouver style as required by PLOS ONE. All references, including those you specifically mentioned (e.g., #31, #88), have been carefully corrected to adhere to the journal's formatting guidelines. This includes:

• Author lists: Correctly formatted to list the first six authors followed by "et al." for works with more than six authors.

• DOI links: All Digital Object Identifiers have been standardised to a clickable URL format (e.g., https://doi.org/10.xxxx...).

• General consistency: Punctuation and citation elements have been checked and corrected for consistency throughout the list.

Thank you once again for your meticulous review and guidance. We are confident that these changes have significantly improved the manuscript.

To facilitate your review, we have provided a detailed list below, showing each outdated reference, highlighted in green, immediately followed by its new, more recent counterpart, highlighted in yellow. This clear visual distinction should make it easy to see the specific changes made.

References

14. Von der Ohe W, Persano Oddo L, Piana ML, Morlot M, Martin P. Harmonized methods of melissopalynology. Apidologie. 2004;35 Suppl 1:S18–S25. https://doi.org/10.1051/apido:2004050

Rezazadeh A, Mehrabian AR, Maleki H, Shakoori Z, Golbaghi NZ, Sharifi T, et al. Evaluation of bee pollen by characterizing its botanical origin, total phenolic content, and microbial load for the formulation of apitherapy products. PLoS One. 2025;20(9):e0327480. https://doi.org/10.1371/journal.pone.0327480.

15. Aparna AR, Rajalakshmi D. Honey—its characteristics, sensory aspects, and applications. Food Rev Int. 1999; 15:455–471. https://doi.org/10.1080/87559129909541199

Bratosin ED, Tit DM, Pasca MB, Purza AL, Bungau G, Marin RC, Radu AF, Gitea D. Physicochemical and sensory evaluation of Romanian monofloral honeys from different supply chains. Foods. 2025;14:2372. https://doi.org/10.3390/foods14132372.

16. Cuevas-Glory LF, Pino JA, Santiago LS, Sauri-Duch E. A review of volatile analytical methods for determining the botanical origin of honey. Food Chem. 2007;103:1032–1043. https://doi.org/10.1016/j.foodchem.2006.07.068

Pattamayutanon P, Angeli S, Thakeow P, Abraham J, Disayathanoowat T, Chantawannakul P. Volatile organic compounds of Thai honeys produced from several floral sources by different honey bee species. PLoS ONE. 2017;12(2):e0172099. https://doi.org/10.1371/journal.pone.0172099.

20. Bogdanov S, Martin P, Lüllmann C. Harmonised methods of the International Honey Commission [Internet]. 2002 [cited 2025 Jul 19]. Available from: https://www.ihc-platform.net/ihcmethods2009.pdf

da Costa IF, Toro MJU. Evaluation of the antioxidant capacity of bioactive compounds and determination of proline in honeys from Pará. J Food Sci Technol. 2021 May;58(5):1900-1908. https://doi.org/10.1007/s13197-020-04701-1.

28. Laallam H. Étude mélissopalynologique, physicochimique et antibactérienne de quelqueséchantillons de miels du Sud Algérien [thesis]. Ouargla: Université de Ouargla; 2018.

Singh I, Singh S. Honey moisture reduction and its quality. J Food Sci Technol. 2018 Oct;55(10):3861–3871. https://doi.org/10.1007/s13197-018-3341-5.

40. Perezarquillue C. Physicochemical attributes and pollen spectrum of some unifloral Spanish honeys. Food Chem. 1995;54:167–172. https://doi.org/10.1016/0308-8146(95)00022-B

Uçurum HÖ, Tepe Ş, Yeşil E, Güney F, Karakuş S, Kolayli S, et al. Characterization of Turkish pine honey according to their geographical origin based on physicochemical parameters and chemometrics. Eur Food Res Technol. 2023;249:1317–1327. https://doi.org/10.1007/s00217-023-04215-y.

41. Sanz ML, Gonzalez M, De Lorenzo C, Sanz J, Martínez-Castro I. A contribution to the differentiation between nectar honey and honeydew honey. Food Chem. 2005;91:313–317. https://doi.org/10.1016/j.foodchem.2004.06.013.

Recklies K, Peukert C, Kölling-Speer I, Speer K. Differentiation of honeydew honeys from blossom honeys and according to their botanical origin by electrical conductivity and phenolic and sugar spectra. Agric Food Chem. 2021;69(4):1329–1347. https://doi.org/10.1021/acs.jafc.0c05311.

43. Terrab A, Recamales AF, Hernanz D, Heredia FJ. Characterisation of Spanish thyme honeys by their physicochemical characteristics and mineral contents. Food Chem. 2004;88:537–542. https://doi.org/10.1016/j.foodchem.2004.01.068.

Silva LR, Sousa A, Taveira M. Characterization of Portuguese honey from Castelo Branco region according to their pollen spectrum, physicochemical characteristics and mineral contents. J Food Sci Technol. 2017 Jul;54(8):2551–2561. https://doi.org/10.1007/s13197-017-2700-y.

57. Crane E. A book of honey. Oxford: Oxford University Press; 1980.

Polatidou K, Nouska C, Tananaki C, Biliaderis CG, Lazaridou A. Physicochemical and rheological characteristics of monofloral honeys—Kinetics of creaming–crystallization. Foods. 2025;14:1835. https://doi.org/10.3390/foods14101835.

59. Laos K, Kirs E, Pall R, Martverk K. The crystallization behaviour of Estonian honeys. Agron Res. 2011;9:427–432. Available from: https://agronomy.emu.ee/wp-content/uploads/2011/12/p09s208.pdf [cited 2025 Jul 19].

Tappi S, Glicerina V, Ragni L, Dettori A, Romani S, Rocculi P. Physical and structural properties of honey crystallized by static and dynamic processes. J Food Eng. 2021;292:110316. https://doi.org/10.1016/j.jfoodeng.2020.110316.

61. Louveaux J. Les abeilles et leurélevage. Paris: OPIDA; 1985.

Romero CA, Sosa N, Vallejos OA, Navarro AS, Yamul DK, Baldi Coronel BM. Physicochemical, microbiological, and sensory properties of stingless bee honey from Argentina. J Apic Res. 2024;64(3):932–943. https://doi.org/10.1080/00218839.2024.2350310.

62. Shugaba A. Analysis of biochemical composition of honey samples from North-East Nigeria. Biochem Anal Biochem. 2012;2:130. https://doi.org/10.4172/2161-1009.1000139.

Piepiórka-Stepuk J, Sterczyńska M, Stachnik M, Pawłowski P. Effects of refrigerated storage on the physicochemical, color and rheological properties of selected honey. Agriculture. 2025;15:1476. https://doi.org/10.3390/agriculture15141476.

63. Council of the European Union. Council Directive 2001/110/EC of 20 December 2001 relating to honey. Off J Eur Communities. 2002;L10:47–52.

Inaudi P, Garzino M, Abollino O, Malandrino M, Giacomino A. Honey: Inorganic composition as possible marker for botanical and geological assignment. Molecules. 2025;30:1466. https://doi.org/10.3390/molecules30071466.

To align the manuscript with the updated references, we have also carefully revised and, where necessary, rewritten certain sections. These changes, including new phrases, sentences, and paragraphs, ensure that the content is fully supported by the most recent literature. For your convenience, all these modifications within the manuscript are highlighted in yellow.

1. Introduction

Old wording

« Determination of floral origin is commonly achieved through melissopalynology [14], which is often complemented by physicochemical and sensory analyses [15]. In addition, volatile compound profiling has been applied as a reliable approach for authentication [16]. »

Rewording

« The botanical origin of honey is primarily determined through melissopalynology, the analysis of pollen grains [14]. This is often complemented by physicochemical and sensory analyses to provide a more comprehensive characterisation [15]. Additionally, volatile compound profiling has emerged as a reliable and precise technique for verifying authenticity [16]. By integrating these methods, a robust and definitive framework is established for authenticating honey and its floral provenance. This multi-faceted approach ensures a complete and accurate determination of the product's origin. »

2. Materials and methods

2.8. Proline content

Old wording

« Proline, the primary amino acid in honey, was quantified using a spectrophotometric method described by Bogdanov et al. [12,20]. »

Rewording

« Proline, the main amino acid in honey, was quantified spectrophotometrically [12, 20]. »

3. Results and discussion

3.2. Moisture as an indicator of stability

Old wording

« Although moisture adjustment (e.g. pasteurisation, dehydration) is sometimes applied, it is controversial [28]. »

Rewording

« Although techniques like pasteurisation or dehydration are sometimes used to adjust honey's moisture, these methods are not always accepted as they can alter the honey's natural state [28]. »

3.3. Acidity profiles and pH balance

Old wording

« ….. and Turkish honeys (16.33 ± 3.00–34.33 ± 13.04 meq/kg) [40]. »

Rewording

« …..and Turkish honeys (8.0–46.89 meq/kg) [40]. »

3.4. Electrical conductivity and mineral signature

Old wording

« Electrical conductivity (EC) is a robust parameter for distinguishing nectar honeys from honeydew honeys, as the latter generally exhibit higher mineral content and thus greater conductivity. Darker honeys also tend to conduct electricity more efficiently »

Rewording

« Electrical conductivity (EC) is a key metric for differentiating between nectar and honeydew honeys. Honeydew honeys typically have a higher mineral content, resulting in a greater EC. Furthermore, darker honeys generally exhibit more efficient electrical conductivity [5,41]. »

Old wording

« Such variability reflects differences in mineral salts, organic acids, and protein concentrations [43], as well as the impact of botanical origin and regional climatic conditions [44]. »

Rewording

« Such variability is a robust indicator, directly influenced by the concentration of dissolved components like mineral salts, organic acids, and proteins [43]. The levels of these constituents are, in turn, determined by the honey’s botanical origin and the specific regional climatic conditions of its production [44], which dictate its overall chemical composition. »

3.8. Sugars and crystallisation potential

Old wording

« Notably, euphorbia (S22) and sweet white mustard (S37) honeys contained more glucose than fructose, a feature also observed in rapeseed and dandelion honeys [57]. »

Rewording

« Notably, euphorbia (S22) and sweet white mustard (S37) honeys contained more glucose than fructose, a feature also observed in rapeseed (Brassica napus) and dandelion (Taraxacum officinale) honeys [57]. »

References #83 and #84 remain unchanged, as they document the characteristics of Polish honeys (Table 5). Literature data are essential for a complete characterization of Polish honeys. For certain honey varieties, some physicochemical parameters are not available in more recent publications. It should be emphasized that the cited results remain valid, and the cited sources are necessary to provide a broader temporal context. Such data illustrate variability and offer a comprehensive overview of the properties of Polish honeys, and are therefore scientifically justified. Since honey composition depends on many factors, presenting results from different time periods better reflects the variability in their composition.

We are immensely grateful for the meticulous feedback from you and the reviewers, which has been invaluable in significantly improving our manuscript. We hope that in its revised form, the manuscript now meets the high standards for publication in PLOS ONE. Thank you for your time and expertise.

Corresponding authors

Prof. Dr. Hocine Allali

Prof. Dr. Radosław Kowalski

Your sincerely,

Prof. Dr. Radosław Kowalski

---

## [Editor Report · Decision Letter 2]

30 Sep 2025

Comparative physicochemical characterization and sensory profiling of western Algerian and Polish honeys

PONE-D-25-40193R2

Dear Dr. Kowalski,

We’re pleased to inform you that your manuscript has been judged scientifically suitable for publication and will be formally accepted for publication once it meets all outstanding technical requirements.

Kind regards,

Md. Asaduzzaman, Ph.D., M. Engg.

Academic Editor

PLOS ONE

---

## [Editor Report · Acceptance letter]

PONE-D-25-40193R2

PLOS ONE

Dear Dr. Kowalski,

I'm pleased to inform you that your manuscript has been deemed suitable for publication in PLOS ONE. Congratulations! Your manuscript is now being handed over to our production team.

Kind regards,

on behalf of

Dr. Md. Asaduzzaman

Academic Editor

PLOS ONE